# Eco-evolutionary dynamics of clonal multicellular life cycles

Vanessa Ress[1,2], Arne Traulsen[1], Yuriy Pichugin[1,3]*

[1]Max Planck Institute for Evolutionary Biology, Plön, Germany; [2]Hamburg Center for Health Economics, University of Hamburg, Hamburg, Germany; [3]Department of Ecology and Evolutionary Biology, Princeton University, Princeton, United States

**Abstract** The evolution of multicellular life cycles is a central process in the course of the emergence of multicellularity. The simplest multicellular life cycle is comprised of the growth of the propagule into a colony and its fragmentation to give rise to new propagules. The majority of theoretical models assume selection among life cycles to be driven by internal properties of multicellular groups, resulting in growth competition. At the same time, the influence of interactions between groups on the evolution of life cycles is rarely even considered. Here, we present a model of colonial life cycle evolution taking into account group interactions. Our work shows that the outcome of evolution could be coexistence between multiple life cycles or that the outcome may depend on the initial state of the population – scenarios impossible without group interactions. At the same time, we found that some results of these simpler models remain relevant: evolutionary stable strategies in our model are restricted to binary fragmentation – the same class of life cycles that contains all evolutionarily optimal life cycles in the model without interactions. Our results demonstrate that while models neglecting interactions can capture short-term dynamics, they fall short in predicting the population-scale picture of evolution.

## Editor's evaluation

This article models the evolution of simple multicellular life cycles using evolutionary game theory. The authors discuss natural selection between different life cycles by modeling growth, fragmentation, and interactions between propagules, discovering conditions for selection of a single life cycle or coexistence of multiple ones. Overall, the model is biologically intuitive, the results are rigorous, and the implications for the evolution of multicellularity are interesting.

**\*For correspondence:**
pichugin@princeton.edu

**Competing interest:** The authors declare that no competing interests exist.

## Introduction

Multicellular organisms are found everywhere. In all major branches of complex multicellularity (animals, plants, fungi, red and brown algae), organisms are formed by cells staying together after cell division – unlike unicellular species, in which cells part their ways before the next division occurs (*Márquez-Zacarías et al., 2021*; *Herron et al., 2022*). However, organisms have to reproduce as otherwise their species will eventually go extinct. For a multicellular organism, this means that some cells must depart in order to develop into an offspring individual. The combination of organism growth and reproduction constitutes a clonal life cycle. Emergence of clonal multicellular life cycles was the central innovation in the earlier stages of the evolution of multicellularity. There, traits, which do not even exist for unicellular species, become crucial for long-term success of even the most primitive colony of cells (*Maynard Smith and Szathmáry, 1995*, *Michod, 2007*). These include the number of cells in the colony, how often cells depart to give rise to new colonies, how large the released propagules are, and how many of them are produced. As the reproduction

and, consequently, fitness of simple cell colonies are dependent on these traits, they immediately become subjected to natural selection, favoring some life cycles over others. Since complex multicellular life descends from those loose cell colonies, the understanding of the prior evolution of primitive life cycles is essential to our understanding of the later evolution of complex traits (*Ratcliff et al., 2012*; *De Monte and Rainey, 2014*; *Hammerschmidt et al., 2014*; *Doulcier et al., 2020*).

There are several theoretical approaches to the modeling of the evolution of multicellular life cycles. The mechanistically simplest class of models assumes that natural selection operates by means of growth competition. Colonies are born small, but due to cell divisions they increase in size and, eventually, fragment, so the number of colonies in the population increases. The life cycle maximizing the population growth rate has a selective advantage as it outgrows all competitors (*Roze and Michod, 2001*; *Libby et al., 2014*; *Pichugin et al., 2017*; *Pichugin et al., 2019*; *Staps et al., 2019*; *Gao et al., 2019*; *Pichugin and Traulsen, 2020*; *Gao et al., 2021*; *Pichugin, 2022*; *Pichugin and Traulsen, 2022*). For groups made of identical cells, growth competition models of evolution predict that some life cycles cannot be the winners of this growth competition under any conditions. For instance, if the fragmentation event is instantaneous and its execution does not cost anything to the group, only fragmentation into two pieces can evolve (*Pichugin et al., 2017*; *Pichugin and Traulsen, 2020*). And indeed, the division into two pieces is, by a large margin, the most common reproductive strategy among microscopic life forms.

However, these models, due to their conceptual simplicity, assume unconstrained (exponential) growth of the population, which cannot be sustained for a prolonged period of time, because resources and space are limited. Other models consider density-dependent growth (*Rossetti et al., 2011*; *Tarnita et al., 2013*; *van Gestel and Nowak, 2016*; *Doulcier et al., 2020*, *Henriques et al., 2021*), where the population growth decreases with the number of groups. A similar approach is the Moran birth–death process on the group level, where whenever a new group emerges, one other group dies (*Traulsen and Nowak, 2006*; *Matias Rodrigues et al., 2012*; *Simon et al., 2013*; *Luo, 2014*; *Kaveh et al., 2016*; *Olejarz et al., 2018*; *Cooney, 2019*; *Cooney, 2020*). While the population dynamics of density-dependent population growth is vastly different from the exponential explosion found in models of unconstrained growth, these two approaches lead to identical results for life cycle evolution: as shown in *Pichugin and Traulsen, 2020*, the dynamics of the fraction of a given life cycle in a population are identical in models with unconstrained and density-dependent growth. Therefore, even in models with density-dependent growth, the evolutionary success of the life cycle is still fully determined by the population growth rates.

Nevertheless, density-dependent growth is also a simplification as different groups may differ in their competitiveness. For instance, large-cell colonies are able to block single cells from access to vital resources (*Rainey and Travisano, 1998*; *Rainey and Rainey, 2003*; *Hammerschmidt et al., 2014*), which may even lead to a complete extinction of solitary cells. Thus, the population dynamics of multicellular life cycles is not necessarily density dependent, but could be frequency dependent – the impact of resource limitation on the population growth depends on both the size and the composition of the population.

Hence, the evolution of multicellular life cycles cannot always be reduced to growth competition, but may arise from eco-evolutionary dynamics.

From a broader empirical perspective, frequency-dependent dynamics is found to be common among microbial populations (*Levin, 1988*; *Ribeck and Lenski, 2015*; *Healey et al., 2016*; *Friedman et al., 2017*). From the perspective of the theoretical ecology, frequency-dependent evolutionary dynamics arising from interactions between diverse population members has also been considered in detail (*May, 1972*; *Wangersky, 1978*; *Bomze, 1983*; *Huang et al., 2015*; *Huang et al., 2017*; *Bunin, 2017*; *Barbier et al., 2018*; *Kotil and Vetsigian, 2018*; *Tarnita, 2018*; *Farahpour et al., 2018*; *Park et al., 2019*; *Park et al., 2020*). The impact of interactions between individuals is recognized in the context of the emergence of aggregative multicellularity, where cells come together to form collectives (*Garcia and De Monte, 2013*; *De Monte and Rainey, 2014*; *Garcia et al., 2015*; *Miele and De Monte, 2021*). However, both empirical and theoretical ecology approaches tend to overlook frequency dependence in the context of clonal life cycles, where the organism's growth in the course of the life cycle may cause a change in its role in interactions (but see an example in *Tverskoi and Gavrilets, 2022* modeling evolution of germ–soma differentiation).

In this article, we developed a model of the evolution of clonal life cycles under frequency-dependent dynamics, implemented in the form of frequency-dependent colony death rate. We focus on three questions:

- What is the population dynamics of a single life cycle?
- What kind of evolutionary outcomes does frequency-dependent selection bring?
- Are there any patterns or constraints among possible evolutionary outcomes that are universal for multiple forms of frequency dependence?

We first address these questions in the context of simpler models with unconstrained growth. In these models, a population performing any life cycle grows exponentially, the competition between life cycles always results in a single one outcompeting all others (which life cycle will be the winner depends only on the growth/death rates but not on the initial composition of population), and finally, the winner always comes from a limited subset of life cycles (in the simplest version of the model with costless fragmentation – it must be a fragmentation mode producing exactly two offspring). We found that including frequency-dependent interactions between organisms performing different life cycles and thus constraining the total population size changes the answers to these questions. First, the population dynamics leads to a stationary state with a finite population size. Second, we found that interactions between groups allow for situations with bistability or coexistence of multiple life cycles – scenarios impossible in the unconstrained growth model. Third, evolutionary stable strategies in our present model always belong to a limited subset of life cycles – the same one containing possible winners of growth competition in models without group interactions. Thus, we found that despite the fundamental differences between our present model and simpler models with unconstrained growth, some of their results have a direct analogy in a much more general eco-evolutionary context considered here.

## Model
### Population dynamics of a single life cycle
We consider a population consisting of cell groups that grow in size and fragment, giving rise to new groups. Cells within a group of size $i$ divide at rate $b_i$, thus a group of size $i$ grows at rate $ib_i$. Groups also die due to both external environmental factors and within-population competition for resources or space. The death rate of groups of size $i$ due to external factors is $d_i$. Frequency-dependent competition is modeled as the death of groups of size $i$ upon encounter with groups of size $j$ at rate $K_{i,j}$ (see *Figure 1A*).

Whenever a group of maturity size $m$ grows to $m + 1$ cells, it immediately fragments. The fragmentation always occurs by the same pattern and determines the life cycle of a population. We represent a fragmentation pattern by $\kappa$ – a partition of a number $m + 1$. For example, the fragmentation pattern of the unicellular life cycle, in which two daughter cells always go apart, is $\kappa = 1 + 1$ (see *Figure 1B*). Other fragmentation patterns correspond to multicellular life cycles. The simplest of them are the life cycles in which groups grow up to two cells, but fragment upon reaching size three. Such a fragmentation can be performed in two ways: either detachment of a single cell, leading to the fragmentation pattern $\kappa = 2 + 1$, or fission into three solitary cells, $\kappa = 1 + 1 + 1$ (see *Figure 1B*). For simplicity, we assume that the cell number does not change during fragmentation (no cell loss), the sum of a fragmentation pattern $\kappa$ is equal to $m + 1$.

If we denote the abundance of cell groups containing $i$ cells as $x_i$, then the dynamics of population is described by a system of differential equations

$$
\begin{aligned}
\frac{dx_1}{dt} &= -b_1 x_1 - d_1 x_1 + m b_m \pi_1(\kappa) x_m - x_1 \sum_{j=1}^{m} k_{1,j} x_j, \\
\left.\frac{dx_i}{dt}\right|_{i>1} &= (i-1)b_{i-1}x_{i-1} - ib_i x_i - d_i x_i + m b_m \pi_i(\kappa) x_m - x_i \sum_{j=1}^{m} k_{1,j} x_j,
\end{aligned}
\tag{1}
$$

where the first two terms $(i - 1)b_{i-1}x_{i-1} - ib_i x_i$ describe the growth of groups – the positive term represents growth from size $i - 1$ to $i$ and the negative term represents growth from $i$ to $i + 1$. The third term $-d_i x_i$ is the environmentally caused death. The term $m b_m \pi_i(\kappa) x_m$ describes the birth of new groups of size $i$ via fragmentation of larger groups, where $\pi_i(\kappa)$ is the number of groups of size $i$

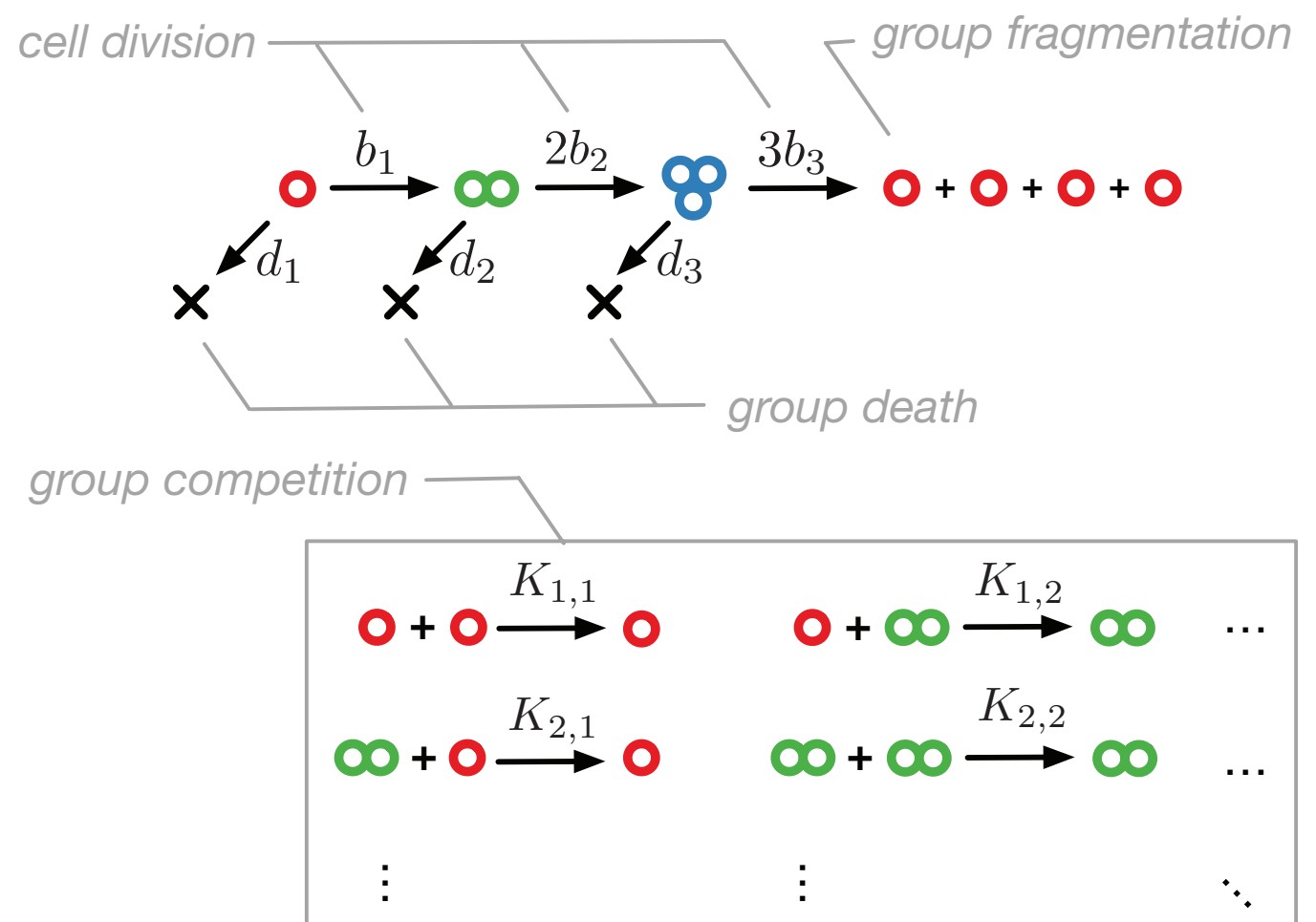

**Figure 1.** Model of clonal life cycles. (**A**) There are four processes occurring in the model: groups grow by cell division, which occurs with rates $b_i$, groups spontaneously die with rates $d_i$, groups fragment immediately upon exceeding maturity size $m$ (3 in this example), according to predefined fragmentation pattern $\kappa$ (1 + 1 + 1 + 1 here), and groups of size $i$ die due to the competition with groups of size $j$ at rates $K_{ij}$. (**B**) Together, group growth and fragmentation constitute a life cycle, in which initially small groups grow in size and eventually fragment, giving rise to more small groups. Colors represent group sizes: red, solitary cells; green, bicellular groups; blue, tricellular groups.

produced in the result of that fragmentation. Finally, $-x_i \sum_{j=1}^{m} K_{i,j} x_j$ is the death of groups due to the competition between groups.

Summarizing the dynamics into matrix notation, the system (1) can be written as

$$\frac{d\mathbf{x}}{dt} = \mathbf{A}\mathbf{x} - \text{diag}(\mathbf{K}\mathbf{x})\mathbf{x}. \tag{2}$$

Here, $\mathbf{x}$ is the column vector of group abundances

$$\mathbf{x} = (x_1, x_2, x_3, \ldots, x_m)^T. \tag{3}$$

The linear term $\mathbf{A}\mathbf{x}$ represents the processes of group growth, fragmentation, and frequency-independent death. The matrix $\mathbf{A}$ of size $m \times m$ is called a population projection matrix in the field of formal demography – in the sense of the projection of the current state into the future state. For an arbitrary life cycle, matrix $\mathbf{A}$ is given by

$$\mathbf{A} = \begin{pmatrix} -b_1 - d_1 & 0 & 0 & \ldots & 0 & mb_m \pi_1(\kappa) \\ b_1 & -2b_2 - d_2 & 0 & \ldots & 0 & mb_m \pi_2(\kappa) \\ 0 & 2b_2 & -3b_3 - d_3 & \ldots & 0 & mb_m \pi_3(\kappa) \\ 0 & 0 & 3b_3 & \ddots & 0 & mb_m \pi_4(\kappa) \\ \vdots & \vdots & \vdots & \ddots & \ddots & \vdots \\ 0 & 0 & 0 & \ldots & (m-1)b_{m-1} & mb_m \pi_m(\kappa) - mb_m - d_m \end{pmatrix}. \tag{4}$$

The elements of the population projection matrix $\mathbf{A}_{i,j}$ represent changes to number of groups of size $i$ due to processes occurring with groups of size $j$ (but not due to interactions). Hence, the population projection matrix has nonzero elements only on the main diagonal (group death and growth of groups to larger sizes), lower subdiagonal (growth of smaller groups), and rightmost column (fragmentation at the end of the life cycle). The elements of the competition matrix $\mathbf{K}$ are given by $K_{i,j}$ for $i,j = 1, \ldots, m$. The operation $\text{diag}(\cdot)$ takes an input vector of length $m$ and transforms it into a diagonal matrix of size $m \times m$ with the entries of the input vector on the main diagonal.

## Population dynamics of multiple life cycles

To investigate the eco-evolutionary dynamics of clonal life cycles, we consider a composition of subpopulations executing various life cycles: $\kappa^{(1)}, \kappa^{(2)}, \ldots \kappa^{(r)}$. In this composite population, the cell growth ($b_i$), environmentally caused (constant) group death ($d_i$), and group fragmentation ($\pi_i(\kappa)$) occur independently in each subpopulation. However, frequency-dependent death due to competition entangles the dynamics of the subpopulations as groups with different life cycles growing together have to compete with each other. If we denote with $x_i^{(j)}$ the number of groups containing $i$ cells in a subpopulation executing the life cycle $\kappa^{(j)}$, the dynamics of the composite population is described by the differential equations

$$\begin{aligned} \frac{dx_1^{(j)}}{dt} &= -b_1 x_1^{(j)} - d_1 x_1^{(j)} + m^{(j)} b_{m^{(j)}} \pi_1(\kappa^{(j)}) x_{m^{(j)}}^{(j)} - x_1^{(j)} \sum_{p=1}^{r} \sum_{k=1}^{n} K_{1,k} x_k^{(p)}, \\ \frac{dx_i^{(j)}}{dt}\bigg|_{i>1} &= (i-1)b_{i-1} x_{i-1}^{(j)} - ib_i x_i^{(j)} - d_i x_i^{(j)} + m^{(j)} b_{m^{(j)}} \pi_i(\kappa^{(j)}) x_{m^{(j)}}^{(j)} - x_i^{(j)} \sum_{p=1}^{r} \sum_{k=1}^{n} K_{i,k} x_k^{(p)}, \end{aligned} \tag{5}$$

where $m^{(j)}$ is the maturity size of the life cycle $\kappa^{(j)}$, and $n = \max(m^{(1)}, m^{(2)}, \ldots, m^{(r)})$ is the maximal maturity size in the composite population. The difference between the one life cycle system (1) and the system of multiple life cycles (5) is in the last term, where groups from every competing subpopulation contribute to frequency-dependent death.

In vector form, the state of the composite population is described by a concatenation of vector states of each subpopulation:

$$\begin{aligned} \tilde{\mathbf{x}} &= (\mathbf{x}^{(1)^T}, \mathbf{x}^{(2)^T}, \ldots, \mathbf{x}^{(r)^T})^T \\ &= (x_1^{(1)}, \ldots, x_n^{(1)}, x_1^{(2)}, \ldots, x_n^{(2)}, \ldots, x_1^{(r)}, \ldots, x_n^{(r)})^T, \end{aligned} \tag{6}$$

where $\tilde{\mathbf{x}}$ is the column vector describing the state of the composite population, $\mathbf{x}^{(j)}$ is the column vector describing $j$th subpopulation in a form (3). Note that the last entries of any $\mathbf{x}^{(j)^T}$ will be zero if $m^{(j)} < n$. The dynamics of the composite population in *Equation (5)* can be represented in the vectorized form, similar to *Equation (2)*:

$$\frac{d\tilde{\mathbf{x}}}{dt} = \tilde{\mathbf{A}}\tilde{\mathbf{x}} - \text{diag}(\tilde{\mathbf{K}}\tilde{\mathbf{x}})\tilde{\mathbf{x}}. \tag{7}$$

Here, the composite population projection matrix representing the cell growth, environmentally caused (constant) group death, and group fragmentation is a diagonal block matrix

$$\tilde{\mathbf{A}} = \begin{pmatrix} \mathbf{A}^{(1)} & 0 & 0 & \dots & 0 \\ 0 & \mathbf{A}^{(2)} & 0 & \dots & 0 \\ 0 & 0 & \mathbf{A}^{(3)} & \dots & 0 \\ \vdots & \vdots & \vdots & \ddots & \vdots \\ 0 & 0 & 0 & \dots & \mathbf{A}^{(r)} \end{pmatrix}, \tag{8}$$

where $\mathbf{A}^{(i)}$ is the population projection matrix of the life cycle $\kappa^{(i)}$ extended to size $n \times n$ ($n$ is the maximal maturity size across all competing life cycles). If the maturity size of the life cycle $i$ is $m^{(i)} = n$, this matrix has a form exactly as in *Equation (4)*. If the maturity size is smaller, $m^{(i)} < n$, then the top-left $m^{(i)} \times m^{(i)}$ has the form (4), while the remaining elements are nonzero at the main diagonal and the lower subdiagonal, as dictated by *Equation (5)*.

The composite competition matrix $\tilde{\mathbf{K}}$ is constructed as

$$\tilde{\mathbf{K}} = \begin{pmatrix} \mathbf{K} & \mathbf{K} & \mathbf{K} & \dots & \mathbf{K} \\ \mathbf{K} & \mathbf{K} & \mathbf{K} & \dots & \mathbf{K} \\ \mathbf{K} & \mathbf{K} & \mathbf{K} & \dots & \mathbf{K} \\ \vdots & \vdots & \vdots & \ddots & \vdots \\ \mathbf{K} & \mathbf{K} & \mathbf{K} & \dots & \mathbf{K} \end{pmatrix}, \tag{9}$$

where each block $\mathbf{K}$ is a competition matrix.

## Invasions from rare

In the general case, the investigation of the composite population dynamics given by *Equation (7)* is a very complex problem without a general solution. Hence, in our study we consider a specific class of initial conditions – invasion from rare, where the composite population contains only two subpopulations: the abundant 'resident' executing life cycle $\kappa^{(R)}$ and rare 'invader' executing different life cycle $\kappa^{(I)}$. This scenario represents an emergence of a mutant in previously stable population of the resident. The population changes if this mutant is capable of invading the resident – otherwise, the mutant goes extinct and the resident population remains the same.

In this scenario, the composite dynamics in *Equation (7)* can be disentangled into resident and invader components. Since the invader population is small, its contribution to frequency-dependent competition is negligible. The members of the resident population compete predominantly between themselves, so the resident dynamics is effectively a single-species scenario,

$$\frac{d\mathbf{x}^{(R)}}{dt} \approx \left[\mathbf{A}^{(R)} - \text{diag}(\mathbf{K}\mathbf{x}^{(R)})\right]\mathbf{x}^{(R)} = \mathbf{0} = \mathbf{A}^{(R,R)}\mathbf{x}^{(R)*}, \tag{10}$$

where the vector $\mathbf{x}^{(R)}$ represents the composition of the resident population, $\mathbf{A}^{(R)}$ is the population projection matrix of the resident, $\mathbf{x}^{(R)*}$ is the equilibrium composition, and we introduced the self-invasion population projection matrix $\mathbf{A}^{(R,R)} = \mathbf{A}^{(R)} - \text{diag}(\mathbf{K}\mathbf{x}^{(R)*})$. Since the resident is assumed to be at a stable equilibrium in the absence of invaders, the self-invasion matrix $\mathbf{A}^{(R,R)}$ has an eigenvalue that is zero, and the equilibrium population composition $\mathbf{x}^{(R)}$ is given by the corresponding eigenvector.

The resident population dynamics can be obtained by solving the nonlinear *Equation (10)*, which in the general case can be performed only numerically.

The rare invader population also competes primarily with the resident and self-competition is negligible. Thus, its dynamics is given by

$$\frac{d\mathbf{x}^{(I)}}{dt} \approx \left[ \mathbf{A}^{(I)} - \mathrm{diag}(\mathbf{K}\mathbf{x}^{(R)*}) \right] \mathbf{x}^{(I)} = \mathbf{A}^{(I,R)}\mathbf{x}^{(I)}, \tag{11}$$

where vector $\mathbf{x}^{(I)}$ represents the composition of the invader population, and we introduced the invasion matrix

$$\mathbf{A}^{(I,R)} = \mathbf{A}^{(I)} - \mathrm{diag}(\mathbf{K}\mathbf{x}^{(R)*}). \tag{12}$$

Unlike the resident dynamics, the dynamics of the invader population is linear – the invasion population projection matrix $\mathbf{A}^{(I,R)}$ is independent from the invader population state $\mathbf{x}^{(I)}$. The linear dynamics of clonal life cycles has been extensively studied in previous work (*Pichugin et al., 2017*). If the largest eigenvalue of the invasion matrix $\mathbf{A}^{(I,R)}$ is positive, then the invader population will increase in numbers, independently of its initial demography. Otherwise, the invader population goes extinct.

The assumption of a negligible impact of the invader population on competition limits the analysis to the early stages of invasion, when the invader population is small. Nevertheless, this makes it possible to investigate the stability of resident life cycles with respect to invasions.

## Results

We first briefly recap the population dynamics and evolution under a more basic model with unconstrained growth ($K_{ij} = 0$) (*Pichugin et al., 2017*; *Pichugin and Traulsen, 2020*). This model has three main features. First, a population executing a single life cycle grows exponentially in the long run. The population growth rate is given by the leading eigenvalue of the population projection matrix ($\mathbf{A}$) and the demographic composition is given by the associated eigenvector. Second, selection always finds a single winner. In a composite population, where different subpopulations execute different life cycles, only one life cycle survives in the long run – the one that has the largest growth rate. This outcome is independent of the initial state of the population. Third, some life cycles cannot be optimal under any combination of cell division rates ($b_i$) and group death rates ($d_i$). In the simplest case of instant and costless group fragmentation, life cycles with more than two offspring cannot win the growth competition.

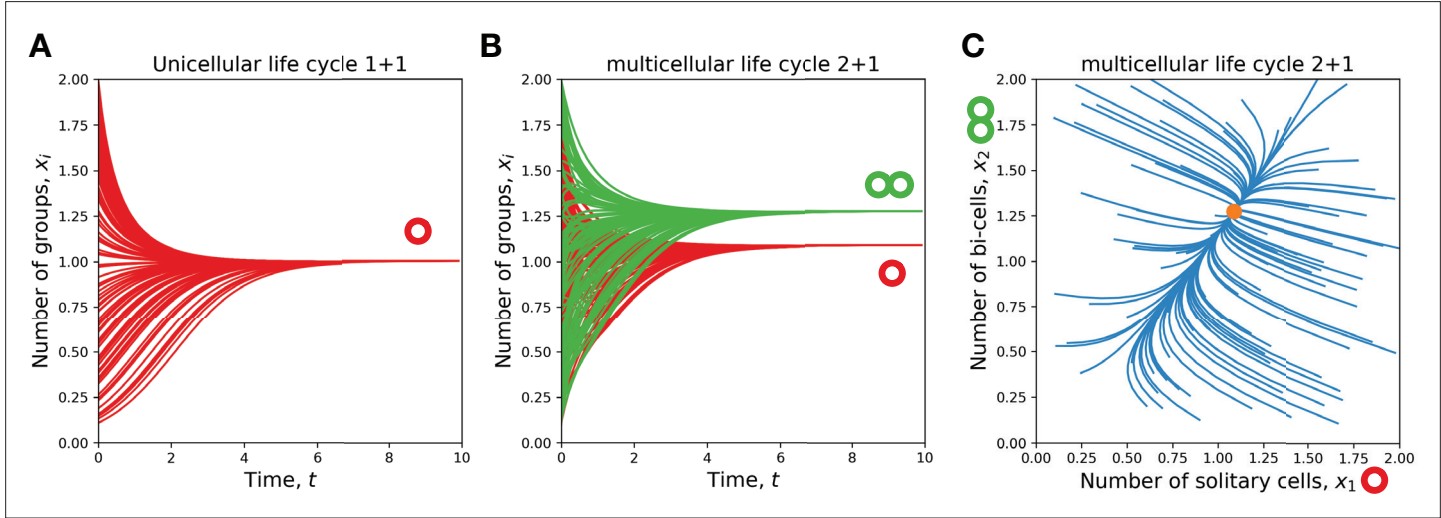

**Figure 2.** A single life cycle comes to a stationary state. (**A**) For a unicellular life cycle $1 + 1$, our model is equivalent to logistic growth. There, the number of solitary cells (red) approaches the carrying capacity from any initial state. (**B, C**) For multicellular life cycles ($2 + 1$ depicted), the population also approaches such a stationary state from any initial state. (**B**) shows the dynamics of solitary cells (red) and bicellular groups (green) with time. (**C**) shows the trajectories of the population composition, the dot marks the stationary state. (**B**) and (**C**) show the same dataset.

Next, we consider how these features transfer to a system taking into account competition between groups. We begin with the dynamics of a single life cycle (section 'Dynamics of a single life cycle') in a system with a population size constraint, which is very different from exponential growth. Then, we consider how the competition proceeds in our model: for two (section 'Competition between pairs of life cycles may result in coexistence or bistability') and multiple life cycles (section 'Competition between multiple life cycles'), where a rich spectrum of possible stationary states is found. After that, we outline the restrictions imposed on evolutionary stable strategies (section 'The set of possible evolutionary stable strategies is restricted'). We conclude with presenting scenarios of a special interest: interactions with killer and victim kernels, where the evolutionary dynamics is reduced to the competition for growth rate and carrying capacity respectively (section 'Killer and victim kernels guarantee a dominance of a single life cycle'), and investigate the competition between unicellular and multicellular life cycles (section 'Conditions promoting the evolution of multicellular life cycles').

## Dynamics of a single life cycle

For the simplest unicellular life cycle ($\kappa = 1 + 1$), where population is composed only of solitary cells, the dynamics of our model given by *Equation (2)* reduces to logistic growth,

$$\frac{dx_1}{dt} = (b_1 - d_1)x_1 - K_{1,1}x_1^2, \tag{13}$$

where the net growth rate is equal to $b_1 - d_1$ and the carrying capacity is $(b_1 - d_1)/K_{1,1}$. The characteristic feature of logistic growth is that the population approaches the carrying capacity with time, starting from either high or low populations size (see *Figure 2A*). The population dynamics of more complex life cycles also bears similarity to the logistic growth of the unicellular life cycle. If a population is small, the competition term is small, so the population grows exponentially. While population size increases, so does the competition term – group mortality rises and the overall population growth slows down. The growth stops when the population reaches a stationary state $\mathbf{x}^*$, where

$$\frac{d\mathbf{x}^*}{dt} = \mathbf{A}\mathbf{x}^* - \mathrm{diag}(\mathbf{K}\mathbf{x}^*)\mathbf{x}^* = \mathbf{0}. \tag{14}$$

Numerical simulations show that a population executing a single life cycle always comes to the same stationary state $\mathbf{x}^*$ from any initial distribution of group sizes (see *Figure 2B and C*).

## Competition between pairs of life cycles may result in coexistence or bistability

A composite population containing several subpopulations executing different life cycles also reaches a stationary state. In the simplest case, only one life cycle survives, while others go extinct. However, we found that the stationary state may contain more than one life cycle (coexistence). Also, the stationary state may depend on the initial state of the population (multistability). Neither of these can occur in the linear model without competition.

To illustrate these effects, we focus on a pair of life cycles ($\kappa^{(1)}, \kappa^{(2)}$) with the special initial conditions, where one of these life cycles is abundant, while the other one is rare. The life cycle $\kappa^{(1)}$ can invade from rare into the abundant $\kappa^{(2)}$ if the largest eigenvalue of the invasion matrix $\mathbf{A}^{(1,2)}$ is positive; see *Equation (12)*. Otherwise, the rare life cycle $\kappa^{(1)}$ goes extinct. For a pair of life cycles, there are

**Table 1.** Competitive interactions in a pair of life cycles can lead to the dominance of either life cycle, their coexistence, or bistability.

| | $\lambda(\mathbf{A}^{(1,2)}) < 0$ <br> $\kappa^{(1)}$ **cannot invade into** $\kappa^{(2)}$ | $\lambda(\mathbf{A}^{(1,2)}) > 0$ <br> $\kappa^{(1)}$ **can invade into** $\kappa^{(2)}$ |
|---|---|---|
| $\lambda(\mathbf{A}^{(2,1)}) < 0$ <br> $\kappa^{(2)}$ cannot invade into $\kappa^{(1)}$ | Bistability between $\kappa^{(1)}$ and $\kappa^{(2)}$ | Dominance of $\kappa^{(1)}$ |
| $\lambda(\mathbf{A}^{(2,1)}) > 0$ <br> $\kappa^{(2)}$ cannot invade into $\kappa^{(1)}$ | Dominance of $\kappa^{(2)}$ | Coexistence of $\kappa^{(1)}$ and $\kappa^{(2)}$ |

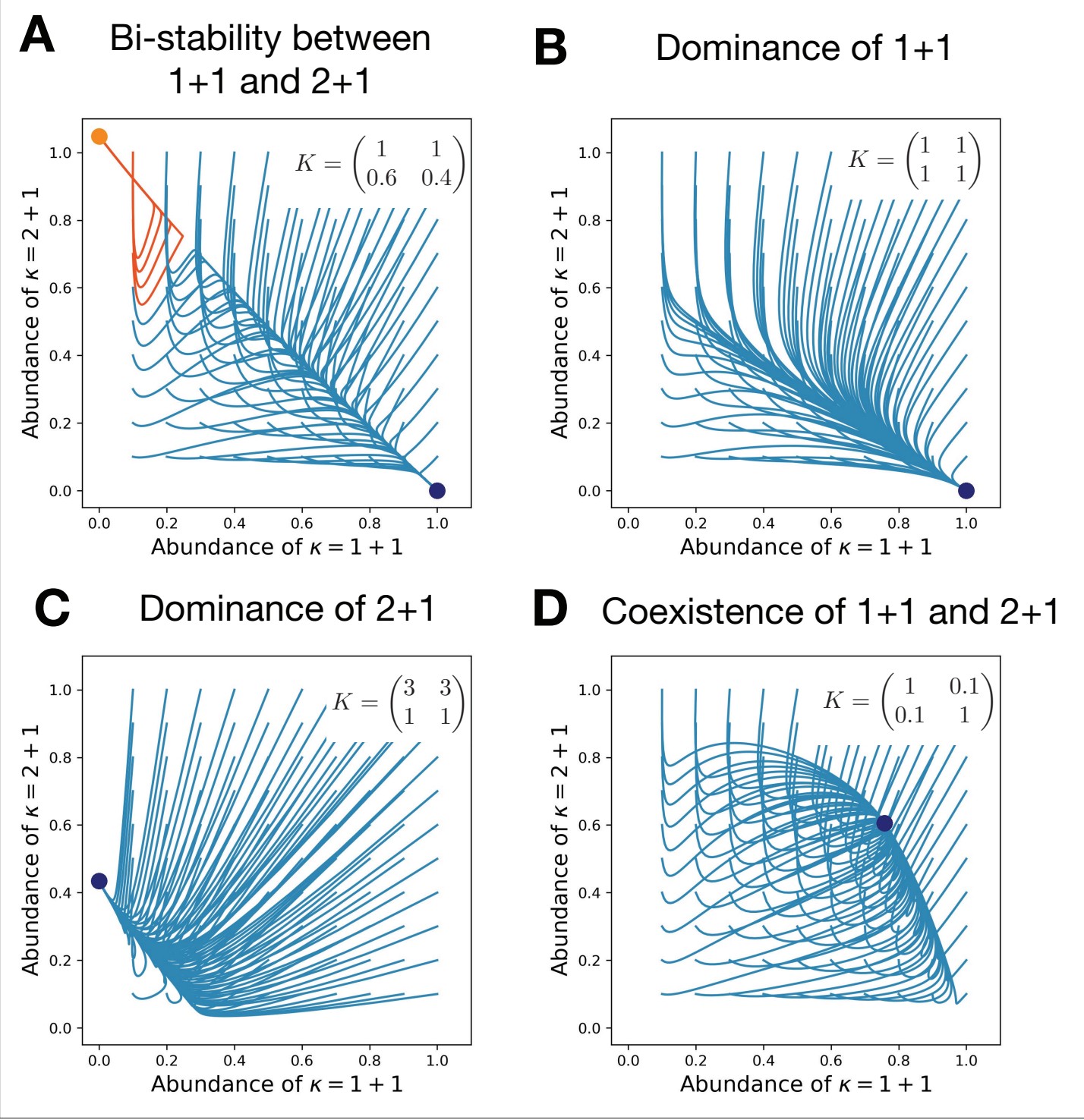

**Figure 3.** Competitive interactions can lead to the bistability between life cycles (**A**), a dominance of either of them (**B,C**), or their coexistence (**D**). Each panel shows the evolutionary trajectories of populations with various initial abundances of the life cycles $1 + 1$ and $2 + 1$ (lines) and all final states (dots). In (**A**), colors indicate to which of two stationary states the trajectory converges. Birth and death rates are $b = (1, 0.5)$ and $d = (0, 0)$; they favor the unicellular life cycle $1 + 1$ in the absence of competition. Simulations performed on each panel only differ in the competition matrix **K** shown in panels.

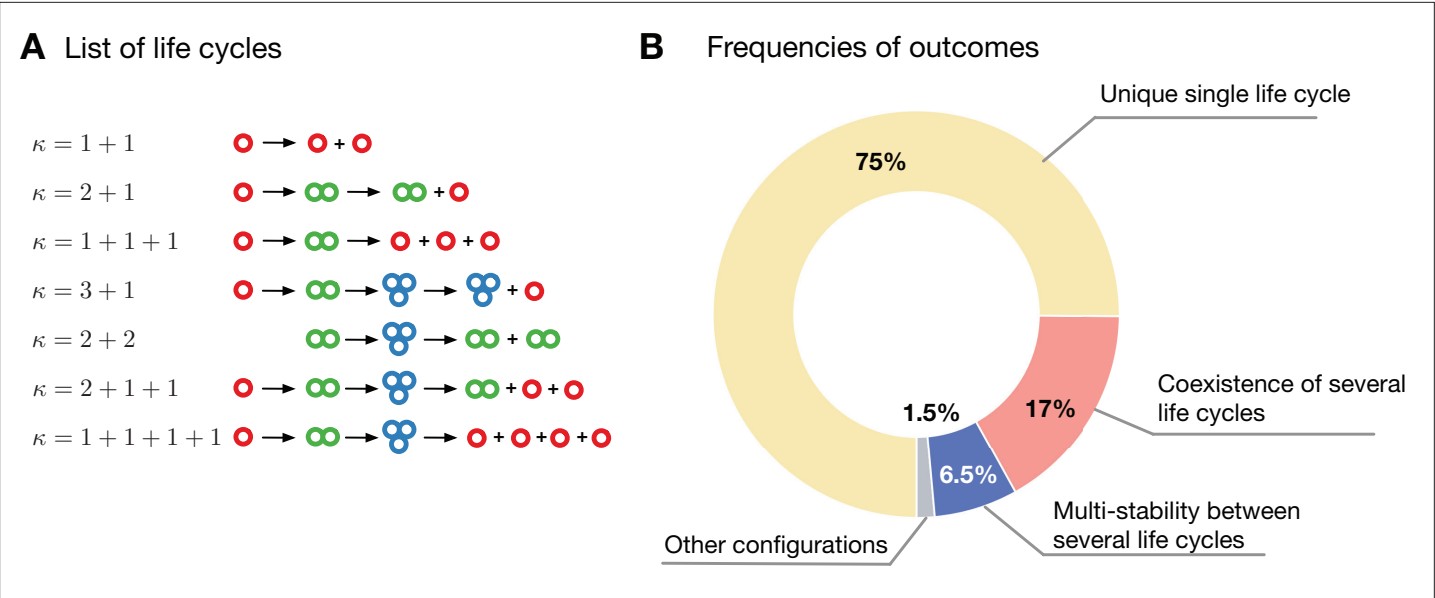

**Figure 4.** Evolution in a population with multiple competing life cycles. (**A**) The schematics of the life cycles taking part in evolution. Groups colored according to the number of cells. (**B**) The frequencies of various classes of outcomes. A single life cycle (yellow) is the most common. Coexistence of life cycles (red) and bistability between them (blue) are less common. More complex composite configurations (gray) are also possible, but found rarely.

four possible scenarios of invasion from rare (see the outline in *Table 1* and corresponding dynamics in *Figure 3*).

Life cycle $\kappa^{(1)}$ dominates life cycle $\kappa^{(2)}$ if $\kappa^{(1)}$ spreads from rare, but $\kappa^{(2)}$ does not. This is equivalent to the leading eigenvalue of the invasion matrix $\mathbf{A}^{(1,2)}$ being positive, while the leading eigenvalue of $\mathbf{A}^{(2,1)}$ is negative. Then, independently of the initial conditions, only the life cycle $\kappa^{(1)}$ survives in the long run (see *Figure 3B*). The opposite signs result in the dominance of $\kappa^{(2)}$ over $\kappa^{(1)}$ (see *Figure 3C*).

Beyond a dominance of one life cycle over another, it is possible that each life cycle is able to spread in the abundance of another. This happens when the leading eigenvalues of both invasion matrices $\mathbf{A}^{(1,2)}$ and $\mathbf{A}^{(2,1)}$ are positive. There, the result of interactions between life cycles is a coexistence of both – an outcome impossible in the model without competition (see *Figure 3D*).

Finally, the leading eigenvalues of both invasion matrices could be negative – then neither of the life cycles can invade into another. Then, the result is a bistability between life cycles, where the outcome of interactions depends on the initial conditions – another result impossible in the model without competition (see *Figure 3A*).

The competition between groups plays a major role in determining which of the four invasion patterns occurs. For instance, it is possible that a life cycle having an advantage in the raw growth rate (i.e., dominating in the unconstrained growth model) is dominated by the result of the competition (see *Figure 3C*), where the life cycle 1 + 1 has a faster growth but 2 + 1 still dominates due to the advantage of multicellular groups in competition.

## Competition between multiple life cycles

Extending our analysis beyond just a pair of life cycles, we considered the evolutionary dynamics in a population with multiple of them. We numerically investigated the evolutionary dynamics in a population containing all life cycles in which groups do not exceed a size of three cells. There are seven such life cycles: unicellular (1 + 1), two with bicellular groups (2 + 1 and 1 + 1 + 1), and four with groups of three cells (3 + 1, 2 + 2, 2 + 1 + 1, and 1 + 1 + 1 + 1) (see *Figure 4A*). We generated a set of 13,000 randomly drawn combinations of growth and competition rates from an exponential distribution with unit rate parameter. For each set, we simulated 100 independent replicates of population dynamics that differ in the initial conditions (abundances and demographic composition of each of the seven life cycles).

These runs were classified by the state of population at the end of simulation (see *Figure 4B* and *Table 2*). The majority of simulations (about 75%) resulted in the survival of a single life cycle across all

**Table 2.** Observation frequencies of different kinds of evolutionary outcomes in a population with multiple competing life cycles.

| Outcome | | Frequency |
|---|---|---|
| Single life cycle | | 0.751 |
| Coexistence of | Two life cycles | 0.121 |
| | Three life cycles | 0.044 |
| | Four or more life cycles | 0.003 |
| Multistability between | Two life cycles | 0.065 |
| | Three life cycles | 0.002 |
| | Four or more life cycles | 0 |
| Composite configurations | Bistability with a coexisting pair | 0.007 |
| | Other configurations | 0.007 |

100 replicates. The next most common outcome is a coexistence between two or more life cycles – found in about 17% of simulations. Also, a multistability between two or three life cycles was observed in about 6.5% of simulations. Here, coexistence describes a situation where the stationary state of the population is composed of the same set of at least two life cycles in every replicate. Multistability describes a situation, where in every replicate the stationary state contained only a single life cycle, but there were different stationary states among the replicates. More complex outcomes were also observed – these were the compositions of multistability and coexistences, which contributed to only about 1.5% of simulations. The most common of these composite situations is a minimal combination of bistability and coexistence: there are two possible stationary states, one is a single life cycle and another is a coexistence of two other life cycles (0.7% of simulations).

The numerical investigation of evolutionary dynamics of multiple life cycles revealed that a diverse range of outcomes are possible, including multistability and coexistence of life cycles, as well as their combinations. At the same time, the most common result is still the dominance of a single life cycle.

## The set of possible evolutionary stable strategies is restricted

Between simple dominance and multistability, about 80% of evolutionary simulations ended with the survival of a single life cycle. This happens if a life cycle is an evolutionary stable strategy (ESS), for example, if it is abundant, it cannot be invaded by any other life cycle. In the basic model without competition, the life cycle with the maximal growth rate also satisfies the definition of an ESS. Here, we found that the set of evolutionary stable strategies is similarly restricted – it also contains only fragmentations into exactly two pieces.

To show this restriction, we consider a triplet of life cycles $\kappa^{(1)}$, $\kappa^{(2)}$, $\kappa^{(3)}$. If the life cycle $\kappa^{(1)}$ is a resident, there are four variants of its stability against invasions: (i) either it is stable against invasions from both $\kappa^{(2)}$ and $\kappa^{(3)}$ (then $\kappa^{(1)}$ is an ESS), or (ii) stable against only $\kappa^{(2)}$, or (iii) stable against only $\kappa^{(3)}$, or (iv) both $\kappa^{(2)}$ and $\kappa^{(3)}$ can invade. Similar four variants exist for the two other life cycles. As a result, for the whole triplet, there are $4^3 = 64$ possible pairwise invasion patterns, which could feature 0, 1, 2, or 3 evolutionary stable strategies.

Numerical simulations show that all 64 patterns can be expressed for the same triplet of life cycles for some combination of cell birth, group death, and competition rates (see *Figure 5A*). We generated a set of 40,000 randomly drawn combinations of these rates from an exponential distribution with unit rate parameter and analyzed the pairwise invasion patterns for each. The six most frequent patterns, comprising 77% of the generated dataset, feature a hierarchical dominance, where the life cycles can be ordered in a way that higher-order life cycle dominates (always invade) lower-order life cycles. These six patterns are all possible hierarchical dominance triplets as there are exactly six ways how three items can be placed in order. If we use the same analysis for the basic, linear model with unrestricted growth, we will only observe hierarchical dominance as larger growth rate results in domination there. On the opposite side of the frequency spectrum, the two most rare patterns feature cyclic dominance, together comprising only 0.015% of the dataset. There, in any pair of life cycles one

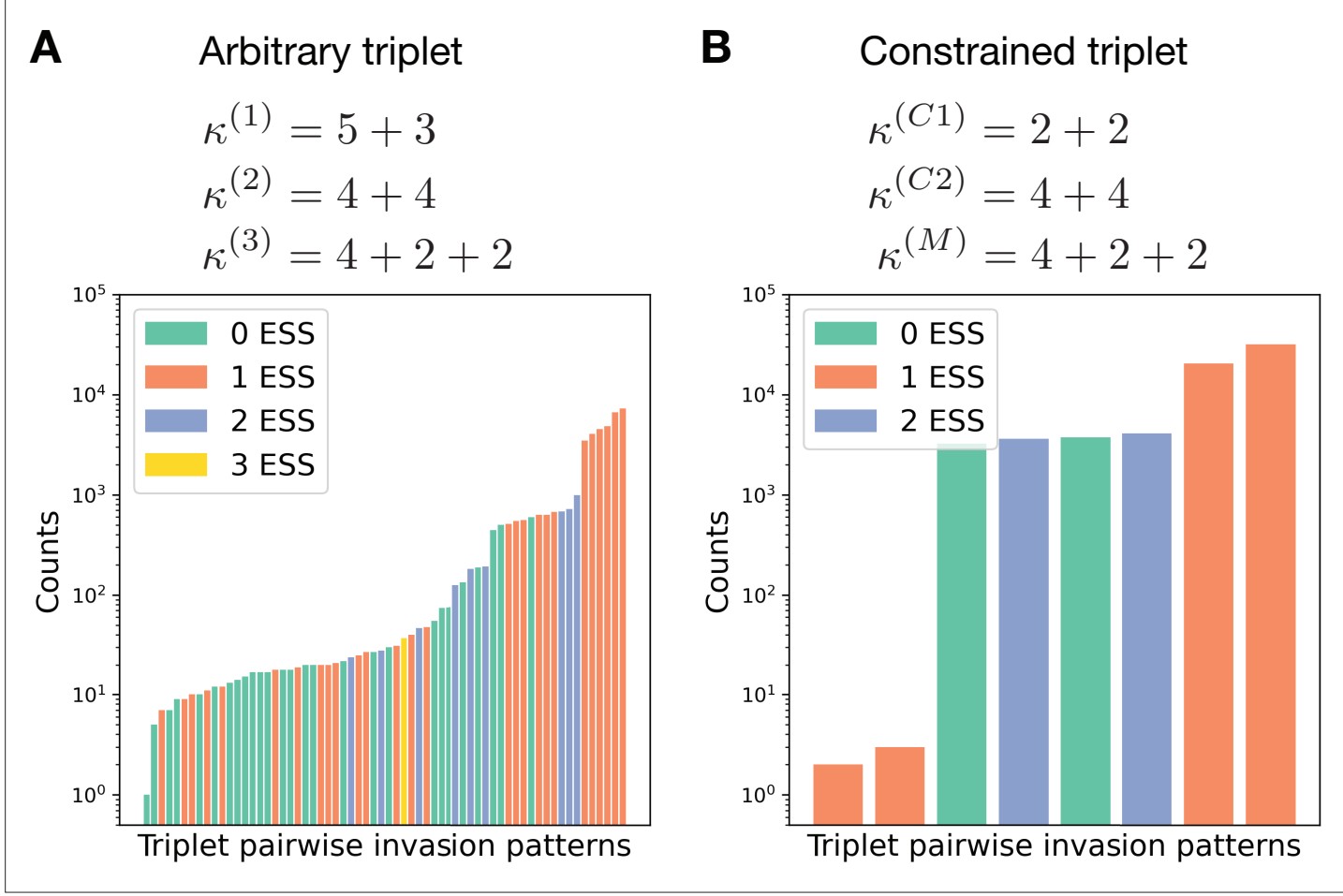

**Figure 5.** Constrained triplets demonstrate fewer patterns of pairwise invasion. (**A**) For a combination of three life cycles, there are $2^{2^3} = 64$ pairwise invasion patterns possible. For the triplet $\kappa^{(1)} = 5+3$, $\kappa^{(2)} = 4+4$, $\kappa^{(3)} = 4+2+2$, the rates of all processes $(b_i, d_i, K_{i,j})$ were randomly sampled from an exponential distribution with unit rate parameter. Then the pairwise invasion pattern was identified. All 64 possible patterns were observed in these simulations. (**B**) In a similar investigation for the triplet $\kappa^{(C1)} = 2+2$, $\kappa^{(C2)} = 4+4$, $\kappa^{(M)} = 4+2+2$, in which $\kappa^{(C1)}$ and $\kappa^{(C2)}$ constrain $\kappa^{(M)}$, only eight patterns were found; see the main text for a discussion.

dominates another but the whole triplet follows a 'rock–paper–scissors' dynamics with no evolutionary stable strategies present (cf. *Park et al., 2020*).

While an arbitrary triplet of life cycles may demonstrate up to 64 invasion patterns, some triplets, which we will call 'constrained,' feature much smaller diversity of patterns. A triplet is constrained if the fragmentation rule of one (constrained) life cycle $\kappa^{(M)}$ can be represented as a combination of fragmentation rules of two other (constraining) life cycles $\kappa^{(C1)}$ and $\kappa^{(C2)}$. The simplest example is the triplet $\kappa^{(C1)} = 2+1$, $\kappa^{(C2)} = 1+1$, $\kappa^{(M)} = 1+1+1$, where the fragmentation of a three-celled group into three solitary cells ($3 \to 1+1+1$) can be presented as a combination of the detachment of a single cell ($3 \to 2+1$) immediately followed by the dissolution of the two-cell group ($2 \to 1+1$). A lot of constrained triplets can be constructed, for example, in our illustrations we use the triplet $\kappa^{(C1)} = 2+2$, $\kappa^{(C2)} = 4+4$, $\kappa^{(M)} = 4+2+2$. Originally, constrained triplets emerged in the model with unconstrained growth (*Pichugin et al., 2017*), where the growth rate of the constrained life cycle $\kappa^{(M)}$ was found to be always between growth rates of the constraining life cycles $\kappa^{(C1)}$ and $\kappa^{(C2)}$. It follows that in the present model with competition ($K_{ij} \neq 0$), each of two constraining life cycles ($\kappa^{(C1)}$ and $\kappa^{(C2)}$) must be either stable against two others or unstable against both. The constrained life cycle $\kappa^{(M)}$ in turn is always invaded by one of constraining life cycles and is stable against the other (see Appendix 1). Hence, the number of possible pairwise invasion patterns for such a triplet is limited to $2 \cdot 2 \cdot 2 = 8$ (see *Figure 5B*). Among these eight patterns, two feature hierarchical dominance, where

the life cycles can be ordered in a way that a higher-order life cycle dominates the lower-order life cycles (with the constrained life cycle $\kappa^{(M)}$ always being in the middle of hierarchy) (see *Figure 6A*). In a larger dataset (66,000 entries) with random birth, death, and competition rates, hierarchical dominance was observed in about 78% of entries. Two patterns feature bistability between constraining life cycles $\kappa^{(C1)}$ and $\kappa^{(C2)}$ (see *Figure 6B*). These patterns appear in about 11% of the dataset. Two more patterns feature a coexistence between all three life cycles (see *Figure 6C*). Note that unlike a pair of life cycles with unique coexistence equilibrium considered in section 'Competition between pairs of life cycles may result in coexistence or bistability,' the triplet features a range of stable coexistence states. The coexistence pattern is similarly frequent, observed in 11% of the dataset. Finally, the least frequent two patterns are non-hierarchical dominance, where one constraining life cycle dominates another, but in the abundance of the constrained life cycle, the invasion pattern is inverse (see *Figure 6D*). They appear with three orders of magnitude lower frequency, smaller than 0.01% of cases.

The fundamental feature of constrained triplets is that any constrained life cycle ($\kappa^{(M)}$) can always be invaded by exactly one constraining life cycles ($\kappa^{(C1)}$ and $\kappa^{(C2)}$). Hence, any life cycle, which can be constrained by two others, cannot be an ESS. We found that any life cycle with more than two offspring can be constrained (see Appendix 1 for the proof). As a result, only binary fragmentation life cycles can be evolutionary stable strategies.

## Killer and victim kernels guarantee a dominance of a single life cycle

As shown above, the evolutionary dynamics of interacting life cycles can be quite complex. In this context, a special interest attracts these cases, where the complexity of dynamics is limited. In this section, we present two forms of interaction matrices (**K**), which guarantee that the evolutionary outcome is a straightforward domination of a single life cycle.

The first special case is the killer kernel, defined as

$$K_{ij} = k_j. \tag{15}$$

There, the probability of a group to die in an encounter depends only on the size of the opponent group ($j$), hence the name.

For an arbitrary killer kernel, a single life cycle has the same demographic composition in the stationary state as it has in the no-interactions model ($K_{ij} = 0$) (see Appendix 2 for the proof).

This feature of the demography leads to the result that the outcome of evolution under a killer kernel is also the same as in the no-interaction model. To show that, consider a composite population containing multiple life cycles; its dynamics is described by *Equation (7)* and depends on the composite projection ($\tilde{\mathbf{A}}$) and competition ($\tilde{\mathbf{K}}$) matrices. If the competition matrix **K** is a killer kernel, the composite competition matrix $\tilde{\mathbf{K}}$ defined in *Equation (9)* is a killer kernel as well. Since the population dynamics of both single and multiple life cycles are governed by equations with the same structure (*Equation (2) and (7)*, respectively), the results of Appendix 2 carry over to a composite population. Specifically, the stationary state of the composite population is proportional to the eigenvector of the composite population projection matrix $\tilde{\mathbf{A}}$ corresponding to its leading eigenvalue. The composite population projection matrix $\tilde{\mathbf{A}}$ defined in *Equation (8)* is a block diagonal matrix, composed of population projection matrices of the life cycles constituting the composite population. Thus, the leading eigenvalue of $\tilde{\mathbf{A}}$ is the largest among the eigenvalues of all population projection matrices comprising $\tilde{\mathbf{A}}$. Additionally, the corresponding eigenvector has nonzero components only at the positions in $\tilde{\mathbf{x}}$ associated with the block having the maximal leading eigenvalue – that is, only that life cycle constitutes the stationary state. This rule is equivalent to the choice of the fastest growing life cycle in the linear model. Thus, the evolution of life cycles competing by a killer kernel ($K_{ij} = k_j$) can be reduced to growth competition, which results in the survival of only one life cycle.

Another special case is the victim kernel defined as

$$K_{ij} = k_i. \tag{16}$$

There, the chance of a group to die depends only on the size of that group ($i$). For an arbitrary victim kernel, the carrying capacity of a single life cycle can be explicitly found. The total number of groups at the stationary state $N^* = \sum_i x_i^*$ is equal to the growth rate of this life cycle in the no-interaction model

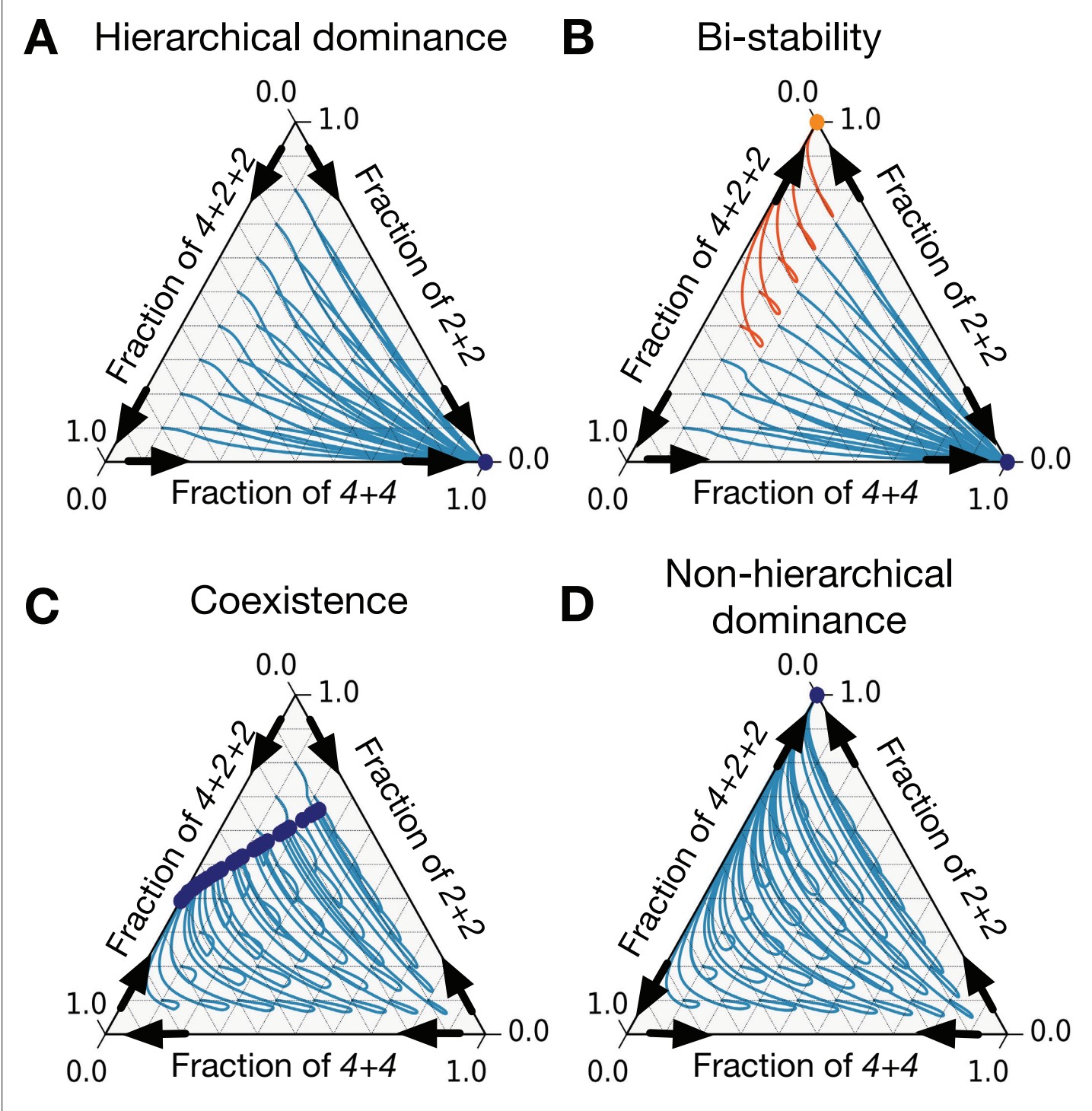

**Figure 6.** There are only eight patterns of pairwise invasion in a constrained triplet of life cycles. Four patterns are shown here and four more are symmetric to them. Blue lines are population dynamics trajectories with different initial composition of the population. Red points are final states. Black arrows show directions of invasion from rare. (**A**) In a hierarchical dominance, one of the constraining life cycles outcompetes the two others from any initial condition. (**B**) In a bistability situation, each of the two constraining life cycles is able to outcompete the third life cycle. Which of them will survive depends on the initial conditions. Trajectories converging to different stationary states are highlighted with different colors. (**C**) In a coexistence, all three life cycles survive to a stationary state. (**D**) In a non-hierarchical dominance, one of the constraining life cycles outcompetes another; however, in the abundance of the constrained life cycle, the dominance is reversed. The parameters used to produce these figures are presented in Appendix 5.

($K_{ij} = 0$) with modified cell division rates $b'_i = b_i/k_i$ and group death rates $d'_i = d_i/k_i$ (see Appendix 3 for the proof).

We found that in the composite population with many life cycles interacting by a victim kernel, only one survives in the long run (see the proof in Appendix 3). In such a scenario, each life cycle grows in numbers if the total population size is smaller than the carrying capacity of that life cycle (derived in Appendix 3). If the whole population size exceeds this value, the life cycle gradually dies out. Hence, selection favors the life cycle with the largest carrying capacity because it can grow in a dense population, when all other life cycles die from intense competition.

In both killer and victim kernels, the evolutionary dynamics is reduced to optimization of a single trait: growth rate for the killer kernel and carrying capacity for the victim kernel. Thus, the result of selection in these scenarios is a dominance of a single life cycle over all suboptimal competitors.

## Conditions promoting the evolution of multicellular life cycles

We conclude our findings with one of the most biologically interesting cases – the evolution of multicellular life cycles in a population dominated by unicellular organisms. This is also one of the simplest scenarios as it deals with the simplest, unicellular life cycle 1 + 1.

If a unicellular population is abundant in the system, its stationary state can be explicitly found (see Appendix 4). Hence, for an arbitrary invading multicellular life cycle $\kappa^{(M)}$, its invasion matrix ($\mathbf{A}^{(M,1)}$) can be found explicitly. As a result, a multicellular invader can spread from rare in a population of unicellular residents if this life cycle has a positive growth rate in a model with unconstrained growth with modified group death rates,

$$d'_i = d_i + \frac{b_1 - d_1}{K_{1,1}} K_{i,1}. \tag{17}$$

The successful multicellular invader drives the unicellular life cycle to extinction if the unicellular life cycle cannot invade from rare. If the unicellular life cycle is an invader, its invasion matrix ($\mathbf{A}^{(1,M)}$) has size $1 \times 1$ and the invasion rate is just equal to its only element. The unicellular life cycle cannot invade into a multicellular resident life cycle if

$$b_1 - d_1 - \sum_{i=1}^{m^{(M)}} K_{1,i} x_i^{(M)*} < 0, \tag{18}$$

where $m^{(M)}$ is the maximal group size in the life cycle $\kappa^{(M)}$, and $x_i^{(M)*}$ is the composition of the population when $\kappa^{(M)}$ is abundant. Derivations of both conditions are presented in Appendix 4.

## Discussion

In this article, we have added an ecological dimension to the problem of life cycle evolution. Specifically, we are interested in three aspects of the eco-evolutionary dynamics: (i) What is a population dynamics of a single life cycle? (ii) What is the evolutionary dynamics of multiple life cycles? (iii) What are the possible constraints on the outcomes of that evolution? All three questions has been extensively studied for the model with unconstrained growth (*Pichugin et al., 2017*), and it is natural to contrast our current findings with these results.

First, the introduction of group competition completely changes the population dynamics of the life cycle growth. Instead of the exponential explosion of the population size occurring in the model with unconstrained population size, the growth in our current model slows down with the population size. Eventually, the population approaches a stationary state with the limited population size and constant composition (see *Figure 2B*).

Second, frequency-dependent selection arising from competition allows diverse outcomes of evolution: the stationary state can feature a coexistence of multiple life cycles or multistability between several possible end states, as shown in *Figure 3*. By contrast, evolution in the model with unconstrained growth always results in a survival of a single life cycle, independently of the initial conditions.

Third, despite all the differences in the population and evolutionary dynamics, the restrictions on the possible evolutionary stable strategies are exactly the same in the models with and without population size constraints (see section 'The set of possible evolutionary stable strategies is restricted').

In both models, an ESS may only feature a fragmentation into exactly two pieces. Any life cycle producing more than two offspring can always be invaded by another life cycle with a smaller number of offspring.

Beyond the scope of our model, life cycles with fragmentation into multiple pieces may become evolutionary stable strategies if fragmentation is costly (e.g., imposes a risk of cell loss). There, binary fragmentation is no longer special as a fragmentation into multiple pieces can win growth rate competition. Nevertheless, constrained triplets of life cycles still exist under a costly fragmentation scenario but the constraining condition is different from the one presented here. There, a life cycle containing two different subsets of offspring with the same combined size is constrained between two life cycles, which use either of these subsets twice (*Pichugin and Traulsen, 2020*). For instance, $\kappa^{(M)} = 2 + 1 + 1$ is constrained as it has different offspring subsets 2 and $1 + 1$ with the same combined size. This life cycle is constrained between life cycles $\kappa^{(C1)} = 2 + 2$ and $\kappa^{(C2)} = 1 + 1 + 1 + 1$, which use one of these subsets twice. Since the scenario of costly fragmentation still contains constrained triplets, life cycles satisfying the rule above, such as $2 + 1 + 1$, cannot be evolutionary stable strategies there. Note that this rule works in the present model too (so there are actually two ways to construct constraining triplets), but it is a weaker condition than the rule presented in section 'The set of possible evolutionary stable strategies is restricted' and does not allow to construct any additional constraining triplets.

In the broad context of the eco-evolutionary dynamics, our dynamical *Equations (2) and (7)* bear a similarity with the generalized Lotka–Volterra (GLV) equations widely used in ecology: both contain a linear growth term and a nonlinear competition term (typically of the second order) balancing out the linear growth. However, our equations are not equivalent to the GLV. In the GLV, individuals corresponding to different elements of the population vector reproduce independently, that is, in our terms, the population projection matrix $\mathbf{A}$ is diagonal. In our model, however, an individual group changes its state in the course of the life cycle and the population projection matrix $\mathbf{A}$ is not diagonal. Of course, if the population projection matrix $\mathbf{A}$ can be diagonalized, we can perform a linear transformation $\mathbf{x} \rightarrow \mathbf{Cy}$ ($\mathbf{C}$ is a matrix) to make the linear term in our model diagonal as in GLV. However, in this case, the interaction term will lose the GLV form of the modification of the growth rate ($-x_i \sum_j K_{ij} x_j$, see *Equation (1)*) and will become a general second-order term instead ($- \sum_{jk} K_{ij} x_j x_k$). Given that our system is not a GLV in disguise, it is surprising how much of the analysis presented here has been performed using the approaches designed to analyze GLV systems.

In this article, we found that the competition plays a major role in the evolution of multicellular life cycles. Our choice of the competition matrix values was driven by theoretical aspects of this article: we either use randomly drawn values for numerical simulations or choose specific forms leading to analytical results. Yet, competition in natural populations is neither random nor fine-tuned to mathematically beautiful outcomes. What might empirical competition matrices in a stage-structured population look like? Unfortunately, the demographics of simple multicellular species is not sufficiently studied experimentally. Still, we can consider an example of emergence of *Pseudomonas fluorescens* colonies, where a competition plays a major role in the population dynamics and evolution (*Rainey and Travisano, 1998*; *Rainey and Rainey, 2003*; *Hammerschmidt et al., 2014*). In a still liquid media, these initially unicellular bacteria evolve a 'glue' production, which causes formation of multicellular aggregates. These aggregates float atop the media, gaining an exclusive access to oxygen. Once the entire surface is covered by continuous bacterial biofilm, the unicellular phenotype living in the body of the media is completely denied the oxygen access and dies out. In the framework of our study, the competition matrix $\mathbf{K}$ is determined by the capability to block oxygen and surface area access. Naturally, the more cells there are in the group, the more stress they apply to others and, at the same time, the more resistant to competition they are. In the limit of small population size, single cells have almost no impact on others ($K_{i,1} \ll 1$) and are the most susceptible to oxygen denial ($K_{1,i} \gg 1$). In opposite limit, an established mat of millions of cells just drives everything else to extinction ($K_{i,j \gg 1} \gg 1$). For arbitrary competitors size, the terms $K_{ij}$ should increase with the size of an opponent group ($K_{ij} < K_{i,j+1}$) but decrease with the size of the focal group ($K_{ij} > K_{i+1,j}$). Similar competition matrices should arise in scenarios where being bigger is better.

## Additional information

### Funding
No external funding was received for this work.

### Author contributions
Vanessa Ress, Software, Formal analysis, Investigation, Methodology, Writing - original draft; Arne Traulsen, Formal analysis, Methodology, Writing - original draft, Project administration; Yuriy Pichugin, Conceptualization, Software, Formal analysis, Supervision, Visualization, Methodology, Writing - original draft

### Author ORCIDs
Arne Traulsen  http://orcid.org/0000-0002-0669-5267
Yuriy Pichugin  http://orcid.org/0000-0003-3078-2499

### Decision letter and Author response
Decision letter https://doi.org/10.7554/eLife.78822.sa1
Author response https://doi.org/10.7554/eLife.78822.sa2

## Additional files

### Supplementary files
• MDAR checklist

### Data availability
The code and data are available from https://github.com/yuriypichugin/Eco-evolutionary-dynamics-life-cycles, (copy archived at swh:1:rev:b7122aeec7b11953867e6b5e588701e5a602276d).

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

## Appendix 1

### Invasion of life cycles and restrictions on ESS

In this section, we show that any resident life cycle with fragmentation into multiple parts can always be invaded by at least one life cycle with a smaller number of offspring.

Consider a resident life cycle $\kappa^{(R)}$, in which more than two offspring groups are produced as a result of fragmentation. The initial dynamics of any life cycle $\kappa^{(I)}$ invading from rare can be described by a linear model with death rates modified as

$$d_i \rightarrow d_i + \sum_j K_{i,j} x_j^{(R)}. \tag{19}$$

The invasion is successful if the leading eigenvalue of the corresponding population projection matrix $\mathbf{A}^{(I,R)}$, defined as in *Equation (11)*, is positive.

In the analysis of the linear model (*Pichugin et al., 2017*), it was shown that if the fragmentation preserves the number of cells (no cell loss), then for any multiple fragmentation life cycle, there exist two constraining life cycles with a smaller number of offspring. For any combination of cell division rates ($b_i$) and group death rates ($d_i$), one of the constraining life cycles has a larger growth rate than the focal multiple fragmentation life cycle, while another has a smaller growth rate.

Now consider the invasion from rare in our present model with competition. The initial invasion rate computed to the leading eigenvalue of the matrix $\mathbf{A}^{(I,R)}$ is equal to the growth rate of the invader life cycle in an environment modified according to *Equation (19)*. Therefore, for any resident population, the invasion rate of the constrained life cycle is always in between the invasion rates of the constraining life cycles. If the resident population is formed by the constrained life cycle itself, its self-invasion rate is zero. Hence, one of the constraining life cycles has a larger invasion rate (i.e., it is positive), while another has a smaller invasion rate (negative). As a result, the constrained life cycle can always be invaded by exactly one of its constraining life cycles. No constrained life cycle can be an ESS. Since any life cycle with more than two offspring is constrained, only binary fragmentation can be an ESS.

To conclude Appendix 1, we consider the resident population formed by a constraining life cycle. If the constrained life cycle has a positive invasion rate, then another constraining life cycle must have a positive invasion rate as well. Alternatively, the constrained life cycle has a negative invasion rate, then another constraining life cycle also has a negative invasion rate. Thus, a constraining resident is either resistant to invasion from both other life cycles in a triplet or can be successfully invaded by both of them.

## Appendix 2

### Population dynamics under the killer kernel

In this section, we show that the demographic distribution of populations in a stationary regime is identical in the linear model ($K_{i,j} = 0$) and under the killer kernel ($K_{i,j} = k_j$).

First, we consider a linear model, where the population dynamics is governed by the population projection matrix $\mathbf{A}$:

$$\frac{d\mathbf{x}}{dt} = \mathbf{A}\mathbf{x}. \tag{20}$$

A number of useful properties of this dynamics comes from the Perron–Frobenius theorem for irreducible non-negative matrices. Here, non-negative matrix means a matrix with all elements greater or equal to zero. Matrix $\mathbf{M}$ is irreducible if its representation by a directed graph, where node $i$ has an edge to node $j$ only if $M_{ij} > 0$, is a strongly connected graph, that is, there is a path from any node to any other node. Since the states $i$ and $j$ represent groups of different sizes, and the population projection matrix $\mathbf{A}$ describes the dynamics of the system, the irreducibility of $\mathbf{A}$ means that group of any given size $i$ can reach any other size $j$ through growth and fragmentation.

However, the population projection matrix $\mathbf{A}$ itself is not non-negative as its main diagonal is negative ($A_{ii} = -ib_i - d_i$). It is not always irreducible as well: if the fragmentation mode produces only multicellular offspring (e.g., $\kappa = 2 + 2$), no group can ever reach unicellular state $j = 1$. Both these limitations can be resolved. First, since the negative elements are located only on the main diagonal, the matrix $\mathbf{A}' = \mathbf{A} + \mathbf{I} \cdot \max(ib_i + d_i)$ is non-negative. The matrix $\mathbf{A}'$ has the same eigenvectors as $\mathbf{A}$, and its eigenvalues ($\lambda'$) relate to eigenvalues of the population projection matrix ($\lambda$) as $\lambda' = \lambda + \max(ib_i + d_i)$. Second, the irreducibility arises only when there are groups sizes smaller than the smallest produced offspring. The number of groups of these sizes will continuously decrease in time: they will grow into larger sizes and spontaneously die. Since fragmentation is not able to resupply these small groups, the population will eventually get rid of them, so these small sizes can be discarded from consideration of the long-term dynamics. Thus, without loss of generality, we can truncate the population projection matrix to take into account only the sizes actually emerging in a life cycle. The resulting modified matrix is non-negative and irreducible, hence, the Perron–Frobenius theorem applies. From this theorem, it immediately follows that for arbitrary birth rates ($b_i$), death rates ($d_i$), and the life cycle executed ($\kappa$), the population projection matrix has a simple leading eigenvalue $\lambda$, its eigenspace is one-dimensional, all components of the eigenvector are positive, and no other eigenvectors of this matrix have all their components positive.

Therefore, in the linear model, the stationary regime is an exponentially growing population (*Pichugin et al., 2017*)

$$\mathbf{x}(t) = \mathbf{w}^* e^{\lambda t}, \tag{21}$$

where $\lambda$ is the leading eigenvalue of the matrix $\mathbf{A}$, and $\mathbf{w}^*$ is the corresponding eigenvector,

$$\mathbf{A}\mathbf{w}^* = \lambda \mathbf{w}^*. \tag{22}$$

Under the killer kernel, the death rates due to competition are the same for groups of all sizes. Hence, following *Equation (2)*, the population dynamics of a life cycle under a killer kernel is

$$\begin{aligned}\frac{d\mathbf{x}}{dt} &= \mathbf{A}\mathbf{x} - \mathrm{diag}(\mathbf{K}\mathbf{x})\mathbf{x} \\ &= \mathbf{A}\mathbf{x} - \mathrm{diag}\left(\sum_j k_j x_j \mathbf{1}\right)\mathbf{x} \\ &= \left(\mathbf{A} - \sum_j k_j x_j \mathbf{I}\right)\mathbf{x},\end{aligned} \tag{23}$$

where $\mathbf{1}$ is the vector of ones, and $\mathbf{I}$ is the identity matrix ($\mathrm{diag}(\mathbf{1}) = \mathbf{I}$). It can be shown by a direct calculation that the vector

$$\mathbf{x}^* = \frac{\lambda}{\sum_i k_i w_i} \mathbf{w}^* \tag{24}$$

is the stationary state of the dynamics (23):

$$
\begin{aligned}
\frac{d\mathbf{x}^*}{dt} &= \mathbf{A}\mathbf{x}^* - \sum_j k_j x_j^* \mathbf{x}^* \\
&= \frac{\lambda}{\sum_i k_i w_i}\mathbf{A}\mathbf{w}^* - \left(\frac{\lambda}{\sum_i k_i w_i}\right)^2 \sum_j k_j w_j \mathbf{w}^* \\
&= \frac{\lambda^2}{\sum_i k_i w_i}\mathbf{w}^* - \frac{\lambda^2}{\sum_i k_i w_i}\mathbf{w}^* \\
&= \mathbf{0}.
\end{aligned}
\tag{25}
$$

Note that while any other eigenvector $\mathbf{w}'$ of the population projection matrix $\mathbf{A}$ can be used in *Equation (25)*, only the leading eigenvector $\mathbf{w}^*$ can represent a biologically meaningful population, as all other eigenvectors have negative elements, which would mean negative number of groups of some size at the stationary state.

Since the stationary state $\mathbf{x}^*$ under the killer kernel is proportional to the vector describing the stationary distribution $\mathbf{w}^*$ in the linear model, the population compositions in both scenarios are the same.

## Appendix 3

### Population dynamics under the victim kernel

#### Dynamics of a single life cycle

In this section, we show that the task of finding the equilibrium population size ($N^* = \sum_i x_i^*$) under the victim kernel ($K_{i,j} = k_i$) is mathematically equivalent to the task of finding the population growth rate in the linear model ($K_{i,j} = 0$) with modified cell birth rates ($b_i \to b_i/k_i$) and group death rates ($d_i \to d_i/k_i$).

In the linear model, the population growth rate is found as the leading eigenvalue of the population projection matrix determined by *Pichugin et al., 2017*

$$
\begin{aligned}
0 \quad &= \det(\mathbf{A} - \lambda\mathbf{I}) \\
&= \begin{vmatrix}
-b_1 - d_1 - \lambda & 0 & 0 & \ldots & mb_m\pi_1(\kappa) \\
b_1 & -2b_2 - d_2 - \lambda & 0 & \ldots & mb_m\pi_2(\kappa) \\
0 & 2b_2 & -3b_3 - d_3 - \lambda & \ldots & mb_m\pi_3(\kappa) \\
\vdots & \vdots & \vdots & \ddots & \vdots \\
0 & 0 & 0 & \ldots & mb_m\pi_m(\kappa) - mb_m - d_m - \lambda
\end{vmatrix}.
\end{aligned}
\tag{26}
$$

Under the victim kernel, the death rate due to the competition depends only on the size of the outcompeted group. Hence, following *Equation (2)*, the stationary state of a life cycle under a victim kernel satisfies

$$
\begin{aligned}
\frac{d\mathbf{x}^*}{dt} \quad &= \mathbf{A}\mathbf{x}^* - \mathrm{diag}(\mathbf{Kx}^*)\mathbf{x}^* \\
&= \mathbf{A}\mathbf{x}^* - \mathrm{diag}\left(\mathbf{k}\sum_j x_j^*\right)\mathbf{x}^* \\
&= \left(\mathbf{A} - N^*\,\mathrm{diag}(\mathbf{k})\right)\mathbf{x}^* \\
&= 0,
\end{aligned}
\tag{27}
$$

where $\mathbf{k}$ is a vector constructed from any column of the competition matrix $\mathbf{K}$ (they are all identical), and $N^* = \sum_i x_i^*$ is the equilibrium population size.

The last equality in *Equation (27)* implies that by Fredholm alternative one out of two conditions is satisfied (*Hoffman, 1971*):

$$
\mathbf{x}^* = 0,
$$

$$
\text{or} \tag{28}
$$

$$
\det(\mathbf{A} - N^*\,\mathrm{diag}(\mathbf{k})) = 0
$$

Limiting ourselves to scenarios where the stationary state $\mathbf{x}^*$ is not an empty population, we can conclude that the population at the equilibrium satisfies

$$
\begin{aligned}
0 &= \det\left(\mathbf{A} - N^* \operatorname{diag}(\mathbf{k})\right) \\[6pt]
&= \begin{vmatrix}
-b_1 - d_1 - k_1 N^* & 0 & 0 & \dots & mb_m \pi_1(\kappa) \\
b_1 & -2b_2 - d_2 - k_2 N^* & 0 & \dots & mb_m \pi_2(\kappa) \\
0 & 2b_2 & -3b_3 - d_3 - k_3 N^* & \dots & mb_m \pi_3(\kappa) \\
\vdots & \vdots & \vdots & \ddots & \vdots \\
0 & 0 & 0 & \dots & mb_m \pi_m(\kappa) - mb_m - d_m - k_m
\end{vmatrix} \\[6pt]
&= k_1 \cdot k_2 \cdot k_3 \cdot \ldots \cdot k_m \\[6pt]
&\quad \cdot \begin{vmatrix}
-\frac{b_1}{k_1} - \frac{d_1}{k_1} - N^* & 0 & 0 & \dots & m\frac{b_m}{k_m}\pi_1(\kappa) \\
\frac{b_1}{k_1} & -2\frac{b_2}{k_2} - \frac{d_2}{k_2} - N^* & 0 & \dots & m\frac{b_m}{k_m}\pi_2(\kappa) \\
0 & 2\frac{b_2}{k_2} & -3\frac{b_3}{k_3} - \frac{d_3}{k_3} - N^* & \dots & m\frac{b_m}{k_m}\pi_3(\kappa) \\
\vdots & \vdots & \vdots & \ddots & \vdots \\
0 & 0 & 0 & \dots & m\frac{b_m}{k_m}\pi_m(\kappa) - m\frac{b_m}{k_m} - \frac{d_m}{k_m} - N^*
\end{vmatrix},
\end{aligned}
\tag{29}
$$

where in the last step, we divided the $i$th column by $k_i$ for all $i$. Comparing the determinants in *Equations (26) and (29)*, we find that they are identical after the substitution

$$
\begin{aligned}
\lambda &\to N^*, \\
b_i &\to \tfrac{b_i}{k_i}, \\
d_i &\to \tfrac{d_i}{k_i}.
\end{aligned}
\tag{30}
$$

Thus, the equilibrium population size ($N^*$) under a victim kernel can be found as a population growth rate in the linear model with modified cell birth and group death rates.

## Competition of multiple life cycles

In this section, we show that in a composite population, in which multiple ($r$) life cycles compete through a victim kernel ($K_{i,j} = k_i$), only one life cycle survives to the stationary state, and this is the same life cycle that has the maximal carrying capacity if grown in isolation, and the same life cycle that would be evolutionarily optimal in the linear model ($K_{i,j} = 0$) with modified cell division ($b_i \to b_i/k_i$) and group death ($d_i \to d_i/k_i$) rates.

If the competition matrix $\mathbf{K}$ constitutes a victim kernel, then the composite competition matrix $\tilde{\mathbf{K}}$ defined as in *Equation (9)* is a victim kernel as well. Hence, our results from section 'Dynamics of a single life cycle' hold also for a composite population. In particular, the population size at the stationary state ($\tilde{N}^* = \sum_{p=1}^{r} \sum_{i=1}^{n} x_i^{(p)}$) is determined by

$$
\begin{aligned}
0 &= \det(\tilde{\mathbf{A}} - \tilde{N}^* \operatorname{diag}(\tilde{\mathbf{k}})) \\
&= \prod_{p=1}^{r} \det(\mathbf{A}^{(p)} - \tilde{N}^* \operatorname{diag}(\mathbf{k})),
\end{aligned}
\tag{31}
$$

where $\tilde{\mathbf{k}}$ is the vector made from a single column of the composite competition matrix $\tilde{\mathbf{K}}$ (it is a concatenation of $r$ vectors $\mathbf{k}$). In the second step of *Equation (31)*, we used the property of block-diagonal matrices: the determinant of a block-diagonal matrix is the product of determinants of its blocks. *Equation (31)* is satisfied if one of the multipliers is equal to zero. The $p$th multiplier becomes zero when the composite population reaches a size equal to the carrying capacity of the $p$th life cycle; cf. *Equation (28)*. At that moment, the $p$th life cycle can neither grow or decay, the life cycles with lower carrying capacity decay as they cannot keep up with competition caused by overcrowding, and only the life cycles with carrying capacity larger than the population size can grow in numbers. *Equation (31)* has $r$ possible solutions with respect to $\tilde{N}^*$ – one for each competing life cycle. However, only one solution can represent a stationary population, where no life cycle can grow in numbers – the one with the maximal population size. There, one life cycle is stationary, while all others decrease in numbers due to overcrowding. Thus, the outcome of life cycle competition under a victim kernel is the survival of a single life cycle, which has the maximal equilibrium population size

among all competitors. According to section 'Dynamics of a single life cycle,' these population sizes are equal to the growth rates of the corresponding life cycles in a linear model with the modified cell division and group death rates. Consequently, the maximal population size corresponds to the fastest growing life cycle in that modified linear model. The population corresponding to the highest eigenvalue takes over and will dominate the system. This means that the life cycle with the largest population size in isolation dominates over all other life cycles in the competition through a victim kernel.

## Appendix 4

### Invasion into unicellular resident and invasion of the unicellular invader

If the resident is unicellular ($\kappa^{(R)} = 1 + 1$), its steady state is given by the solution of

$$(b_1 - d_1)x_1^* - K_{1,1}x_1^{*2} = 0, \tag{32}$$

equal to

$$x_1^* = \frac{b_1 - d_1}{K_{1,1}}. \tag{33}$$

Then, for an arbitrary invader multicellular life cycle $\kappa^{(M)}$, the invasion matrix is given by

$$
\begin{aligned}
\mathbf{A}^{(M,R)} \quad &= \mathbf{A}^{(M)} - \mathrm{diag}(\mathbf{K}\mathbf{x}^{(R)}) \\
&= \mathbf{A}^{(M)} - \mathrm{diag}(K_{i,1}x_1^*) \\
&= \mathbf{A}^{(M)} - \tfrac{b_1 - d_1}{K_{1,1}}\,\mathrm{diag}\left(K_{i,1}\right).
\end{aligned} \tag{34}
$$

This is equivalent to the linear growth of the invader life cycle in an environment with modified death rates

$$d_i' = d_i + \frac{b_1 - d_1}{K_{1,1}}K_{i,1}. \tag{35}$$

If the invader is unicellular ($\kappa^{(I)} = 1 + 1$), then the invasion matrix is effectively a $1 \times 1$ matrix since the invader contains only isolated cells. Even if $\mathbf{A}^{(I,M)}$ formally has a larger size, it is a block matrix, with the element $\mathbf{A}_{1,1}^{(I,M)}$ being a block, with a value that is equal to the growth rate of the unicellular life cycle. This value is

$$
\begin{aligned}
\mathbf{A}_{1,1}^{(I,M)} \quad &= \mathbf{A}_{1,1}^{(I)} - \mathrm{diag}(\mathbf{K}\mathbf{x}^{(M)*})_{1,1} \\
&= b_1 - d_1 - \sum_{i=1}^{m^{(M)}} K_{1,i}x_i^{(M)*},
\end{aligned} \tag{36}
$$

where $x_i^{(M)}$ is the number of groups of size $i$ in the resident population, and $m^{(M)}$ is the maximal group size of the resident life cycle. The unicellular invader cannot spread in a resident population, when $\mathbf{A}_{1,1}^{(I,M)} < 0$.

## Appendix 5

## Parameters of calculation used in figures

### Figure 2

In *Figure 2A*, we used $b_1 = 1, d_1 = 0, K_{11} = 1$. The initial population size was drawn from the uniform distribution $U(0.1, 2)$.

In *Figure 2B*, we used $b = (1, 1)$ and $d = (0, 0)$. The competition matrix was

$$K = \begin{pmatrix} 1 & 0.2 \\ 0.2 & 0.5 \end{pmatrix}.$$

The initial number of groups of each size was randomly drawn from the uniform distribution $U(0.1, 2)$.

### Figure 3

In *Figure 3*, we considered the life cycles 1 + 1 and 2 + 1. We used $b = (1, 0.5)$ and $d = (0, 0)$. For each plotted trajectory, the populations were initialized with $\mathbf{x}_{1+1} = (s_1)$, $\mathbf{x}_{2+1} = (s_2, 0)$, where $s_1, s_2 \in \{0.1, 0.2, 0.3, \dots, 1.0\}$. The dynamics shown in the four panels differ by the competition matrix used:

- Panel A

$$K = \begin{pmatrix} 1 & 1 \\ 1 & 1 \end{pmatrix}$$

- Panel B

$$K = \begin{pmatrix} 3 & 3 \\ 1 & 1 \end{pmatrix}$$

- Panel C

$$K = \begin{pmatrix} 1 & 0.1 \\ 0.1 & 1 \end{pmatrix}$$

- Panel D

$$K = \begin{pmatrix} 1 & 1 \\ 0.6 & 0.4 \end{pmatrix}$$

### Figure 6

In *Figure 6*, we considered the life cycles $\kappa^{(C1)} = 2 + 2$, $\kappa^{(C2)} = 4 + 4$, $\kappa^{(M)} = 4 + 2 + 2$. Panels differ in birth, death, and competition rates. Trajectories in each panel have different initial states. For each initial state, the composite population contains all three life cycles with different fractions $s^{(C1)}, s^{(C2)}, s^{(M)}$, such that $s^{(C1)} + s^{(C2)} + s^{(M)} = 1$. The initial group sizes distribution is proportional to the equilibrium population of that life cycle alone:

$$\tilde{\mathbf{x}}_{t=0} = (s^{(C1)}\mathbf{x}^{*(\mathbf{C1})^T}, s^{(C2)}\mathbf{x}^{*(\mathbf{C2})^T}, s^{(M)}\mathbf{x}^{*(\mathbf{M})^T})^T, \tag{37}$$

where the vectors $\mathbf{x}^{*(\mathbf{C1})^T}$, $\mathbf{x}^{*(\mathbf{C2})^T}$, $\mathbf{x}^{*(\mathbf{M})^T}$ are equilibrium population states of corresponding life cycles grown in isolation, computed according to *Equation (14)*.

- In panel A (hierarchic dominance), we used $b_i = 1.0$, $d_i = 0$ and the competition matrix $K_{ij} = 0.1$:

$$K_{\text{Hier. dom.}} = \begin{pmatrix} 0.1 & 0.1 & 0.1 & 0.1 & 0.1 & 0.1 & 0.1 \\ 0.1 & 0.1 & 0.1 & 0.1 & 0.1 & 0.1 & 0.1 \\ 0.1 & 0.1 & 0.1 & 0.1 & 0.1 & 0.1 & 0.1 \\ 0.1 & 0.1 & 0.1 & 0.1 & 0.1 & 0.1 & 0.1 \\ 0.1 & 0.1 & 0.1 & 0.1 & 0.1 & 0.1 & 0.1 \\ 0.1 & 0.1 & 0.1 & 0.1 & 0.1 & 0.1 & 0.1 \\ 1.1 & 1.1 & 1.1 & 1.1 & 1.1 & 1.1 & 1.1 \end{pmatrix}, \tag{38}$$

or equivalently

$$K_{ij} = \begin{cases} 1.1, & i = 7 \\ 0.1, & \text{otherwise} \end{cases} \tag{39}$$

- In panel B (bistability), we used $b_i = 1$ and $d_i = 0$, while the competition matrix was

$$K_{\text{Bi-stability}} = \begin{pmatrix} 0.1 & 0.1 & 0.1 & 0.1 & 0.1 & 0.1 & 0.1 \\ 0.1 & 0.1 & 0.1 & 0.1 & 0.1 & 0.1 & 10 \\ 0.1 & 0.1 & 0.1 & 0.1 & 0.1 & 0.1 & 0.1 \\ 0.1 & 0.1 & 0.1 & 0.1 & 0.1 & 0.1 & 0.1 \\ 0.1 & 0.1 & 0.1 & 0.1 & 0.1 & 0.1 & 0.1 \\ 0.1 & 0.1 & 0.1 & 0.1 & 0.1 & 0.1 & 0.1 \\ 0.1 & 10 & 0.1 & 0.1 & 0.1 & 0.1 & 0.1 \end{pmatrix}, \tag{40}$$

or equivalently

$$K_{ij} = \begin{cases} 10, & (i,j) = (2,7) \text{ or } (7,2) \\ 0.1, & \text{otherwise} \end{cases} \tag{41}$$

- In panel C (coexistence), we used $b_i = 1$ and $d_i = 0$, while the competition matrix was

$$K_{\text{Coex.}} = \begin{pmatrix} 0.1 & 0.1 & 0.1 & 0.1 & 0.1 & 0.1 & 0 \\ 0.1 & 0.1 & 0.1 & 0.1 & 0.1 & 0 & 0.1 \\ 0.1 & 0.1 & 0.1 & 0.1 & 0 & 0.1 & 0.1 \\ 0.1 & 0.1 & 0.1 & 0 & 0.1 & 0.1 & 0.1 \\ 0.1 & 0.1 & 0 & 0.1 & 0.1 & 0.1 & 0.1 \\ 0.1 & 0 & 0.1 & 0.1 & 0.1 & 0.1 & 0.1 \\ 0 & 0.1 & 0.1 & 0.1 & 0.1 & 0.1 & 0.1 \end{pmatrix}, \tag{42}$$

or equivalently

$$K_{ij} = \begin{cases} 0, & i+j = 8 \\ 0.1, & \text{otherwise} \end{cases} \tag{43}$$

- In panel D (non-hierarchical dominance), we used $d_i = 0$,

$$b_{\text{Nonh. dom.}} = (6.48076, 2.79693, 2.3088, 5.34057, 1.0, 1.62478, 1.32615) \tag{44}$$

and

$$K_{\text{Nonh. dom.}} = \begin{pmatrix} 0.12491 & 0.16453 & 0.40972 & 0.13981 & 0.03496 & 0.79364 & 0.06097 \\ 0.36477 & 0.10859 & 0.09099 & 0.70391 & 0.01714 & 0.53354 & 0.49778 \\ 0.16432 & 0.58285 & 1 & 0.01918 & 0.01268 & 0.08071 & 0.11208 \\ 0.13519 & 0.56771 & 0.11879 & 0.02601 & 0.08905 & 0.1172 & 0.14661 \\ 0.01251 & 0.31353 & 0.02639 & 0.07433 & 0.05312 & 0.22877 & 0.14841 \\ 0.06258 & 0.43833 & 0.30679 & 0.3323 & 0.01014 & 0.09637 & 0.24751 \\ 0.19945 & 0.022 & 0.00087 & 0.2469 & 0.09733 & 0.08247 & 0.37168 \end{pmatrix}. \tag{45}$$

