## [Editor Report]

This article models the evolution of simple multicellular life cycles using evolutionary game theory. The authors discuss natural selection between different life cycles by modeling growth, fragmentation, and interactions between propagules, discovering conditions for selection of a single life cycle or coexistence of multiple ones. Overall, the model is biologically intuitive, the results are rigorous, and the implications for the evolution of multicellularity are interesting.

---

## [Decision Letter]

**Decision letter after peer review:**

Thank you for submitting your article "Eco-evolutionary dynamics of clonal multicellular life cycles" for consideration by *eLife*. Your article has been reviewed by 3 peer reviewers, and the evaluation has been overseen by a Reviewing Editor and Aleksandra Walczak as the Senior Editor. The following individual involved in review of your submission has agreed to reveal their identity: Denis Tverskoi (Reviewer #3).

The reviewers have discussed their reviews with one another, and the Reviewing Editor has drafted this to help you prepare a revised submission. Overall, the reviewers found the manuscript informative and interesting, but have the following suggestions to improve the presentation and clarity of the manuscript. In addition to the consolidated essential revisions listed below, we would highly recommend that you address all the reviewers' comments in your revision.

Essential revisions:

1) Please comment on the impact of the mechanism of interaction and discuss the forms of interactions that might be more relevant in the modeling. In the discussions, it would also be helpful to mention real biological systems that the models represent.

2) Please revise the terminology (e,g, "projection matrix", "linear dynamics", and stationary state") to avoid confusion with technical math definitions.

3) In the discussion "Similarities between models with and without group competition," please include results from models without competition as well to make the comparisons clearer for the reader.

4) Please include additional quantitative discussions and elaborate on the effect of kernel on the outcomes.

5) Please highlight the primary takeaways of the paper among many results that are derived to underscore the main findings. The reviewers suggest that you frame these questions at the beginning of your paper to help your reader follow the modeling and results.

6) Please reorganize the Discussion section to include a recap and improve its flow for your readers.

7) Please double-check the equations for accuracy and address the concerns raised by reviewer #3.

*Reviewer #1 (Recommendations for the authors):*

I have few comments and suggestions to improve the quality of the work:

1. The model is very clearly stated and a connection between current model and previous works has been established. The generalization of interaction between different groups are quite interesting and significant improvement in the modeling approach to the evolution of multicellular life cycles. However, it would be more instructive if the authors identify what form of interactions might be more relevant in the modeling, beside victim and killer. Also, it would be pedagogical if the above is compared, in a couple of sentences or a paragraph, with other frameworks in evolutionary game theory. To me, the model in the absence of interactions is a quasi-species equation where each group is a trait and every growth terms is similar to mutation to a new trait. A given life cycle is a mutational pathway in this analogy. The interaction term is interaction between the different types/traits. It would be instructive if authors, by making such connections with theoretical evolution literature, discuss what qualitative or qualitative changes they already expect from introducing the interactions kernel.

2. The model has a complex form, inevitably. There are several mechanisms interplaying with each other. Growth rates of groups, fragmentation patterns, interaction between groups of the same life cycle, and interaction between groups belonging to different life cycles and most often the latter two are indistinguishable. Even though the model is complex, the paper is well written and the results are clearly enough stated. However, it would be very important to improve the significance of the work by highlighting a few main messages of the paper among many results that are derived to highlight the main findings. This is the part I found somewhat unclear in the current manuscript.

3. Regarding the above comment, I want to suggest that the authors highlight, from the beginning, what questions they are trying to answer and what quantities can be calculated from the model. In principle, I expect the model says the condition for selection of a life-cycle against other ones (invasion of rare) and how it relates to model parameters. Similarly conditions for coexistence between different life cycles. Finally, the steady state values of each group abundances. These are all discussed in the paper to some extent. I just suggest highlighting the main quantities you are calculating and questions you are aiming to answer a little clearer.

4. In the case of two competing life cycles, it is not clear, to me, what is the qualitative result. The kernel is introduced in four forms suggested by the authors (Figure 2). It is concluded that depending on whether one has a killer kernel or victim kernel one life cycle wins and in other cases there might be coexistence. Are there further results? For example can one in principle states the condition for ESS or selection advantage of one life cycle?

5. (Similar to the previous comment) How does different kernels (victim or killer) improve or modify condition for selection of one life cycle while competing with another? The answer is detailed but it is useful to sum up the overall behavior cycles.

6. I want to comment that the three life cycle competition results was interesting and the analogy with rock-paper-scissor game was clarifying the findings.

7. Does the introduction of kernel term overall promotes coexistence between two life cycles?

8. While the paper has a theory approach and the generality of results justifies it, it would be very useful to connect the general finding with the observations in experimental evolution of multicellularity. For example, discuss cases where there are unique life cycles and when there are more than one.

The evolution of multicellular life cycles is a central process in the course of the emergence of multicellularity. The model suggested by authors connects evolution of multicellular life cycles to evolutionary game theory. The introduction of the interaction terms seems to be a great modeling way to discuss under what circumstances different life cycles can coexist or when one life cycle is chosen among other potential ones due to a natural selection among life cycles. The results are discussed in some details but due to complexity in some cases examples are used. I recommend this work for publication in Journal *eLife* after revision. It is a new model with high impact on the field of evolution of multicellular structures.

*Reviewer #2 (Recommendations for the authors):*

1) The first paragraph does not contain any references to back up the claims presented by the authors. It would be good to address this and add appropriate references.

2) In Figure 1 it would be good to explicitly state in the caption what the different colors for the groups mean.

3) In the discussion "Similarities between models with and without group competition "it would be good to explicitly refer to results from results from models without competition (which I think are only seen in Figure 2A) to make the comparisons clearer for the reader.

*Reviewer #3 (Recommendations for the authors):*

Please find my detailed recommendations for the authors below

Page 7, line 142: “The projection matrix $A$…” It is not clear to me what the authors mean by “projection matrix”. From a mathematical point of view, a square matrix $P$ is called a projection matrix if it is equal to its square, i.e. if $P^2=P$. However, it does not seem that in general $A^2 = A$.

Page 10, line 219: “Unlike the resident dynamics, the dynamics of the invader population is linear…” This dynamics is linear only if $x^{(R)}$ is constant (e.g., at equilibrium). Therefore, it would be helpful to use different notations for the function $x^{(R)}(t)$ and its value at the equilibrium $x^{(R,*)}$ in formulas 11-12 describing these dynamics.

Page 11, lines 245-246: “Numerical simulations show that an isolated life cycle always comes to the same stationary state $x^*$ from any initial distribution of group sizes.” Could you please add more details about these numerical simulations? What kind of life cycles were considered? What birth and death rates have been chosen? How was the competition matrix generated?

Throughout the text, the authors discuss the results on various special types of a competition matrix (constant, killer kernel, victim kernel). These special cases are used to obtain analytical conclusions that cannot be drawn in a general case. This helps readers deepen their intuition about this model. However, I wonder if there are any real biological systems that can be described by these special kernels? If so, it would be very helpful to include the corresponding discussion in the text.

Page 25, line 515: “In the linear model, the stationary state is an exponentially growing population…” From a mathematical point of view, a stationary state of a system is a state with all observables independent of time. Therefore, I am not sure if it is appropriate to use the term “stationary state” here.

Page 25, lines 526-529. It is not clear what does the index $s$ mean in the corresponding formulas.

Appendix A1. As was shown in the previous work, the solution to Equation (14) depends on the leading eigenvalue $\λ^*$ of the matrix $A$ and the corresponding eigenvector $w^*$. After refreshing that, the authors showed that a stationary state of the dynamics governed by Formula (21) (i.e., by Equation (14) in the case of the killer kernel) depends on $\λ^*$ and $w^*$ (see Formula (22)). This is correct. However, for each eigenvalue $\λ$ and each corresponding eigenvector $w$, Formula (22) produces a stationary state of the dynamics governed by Formula (21). It is not clear to me why, in the case of the killer kernel, the authors consider only the leading eigenvector? If such a conclusion is made on the basis of the above results for the linear case, please explain why these results can be generalized to the killer kernel.

And the last question. Is it possible that the leading eigenvalue $\λ$ has an eigenspace of dimension higher than 1 so that stationary states (22) form a “line” of equilibria?

Page 25, line 528. I guess, it should be $x_{s,j}^*$ instead of $x_{s,j}$.

Page 27, line 551-552. What is $N_0$? How is it related to $N^*$?

Section 3.2. The authors state that the dynamics are very complex in the case of competition between multiple life cycles, and therefore consider these dynamics only in some special cases. I agree that the dynamics are complex. However, as a reader, I have no intuition about how the model works in the general case, which is the most interesting question for me. Therefore, it would be helpful to add some numerical simulations exploring the above dynamics in the general case. It would also be useful to present some statistics illustrating the average number of different life cycles presented in a stationary state as well as the number of extinct ones. Maybe some life cycles are rarely observed in a stationary state, while others are widespread under a broad range of parameters and initial conditions?

Page 16, Figure 3D. If possible, could you please mark in different colours trajectories approaching different stationary states?

Lines 450-451. “Yet, it is possible to introduce a linear transformation $x \rightarrow Cy$, where $C$ is a matrix, which will make the linear term in our model diagonal.” Where is a proof that the matrix $A$ can be diagonalized?

I think the structure of the Discussion section could be improved. For example, I do not understand why you put paragraph 2 (lines 431-442) at the beginning of the Discussion? This paragraph is about a generalization of your model and its results, and it is strange to discuss the generalization of the results before discussing the results themselves. Second, it might be helpful to add a brief recap of the problem under study, and a brief overview of the model at the beginning of the Discussion section before discussing the position of your study in various contexts, and related results (lines 443-505).

---

## [Author Response]

Essential revisions:1) Please comment on the impact of the mechanism of interaction and discuss the forms of interactions that might be more relevant in the modeling. In the discussions, it would also be helpful to mention real biological systems that the models represent.

We have now described the interactions in more detail and extended the discussion. Concerning the biology, we draw inspiration from several microbial systems, but we do not make detailed predictions for any particular system – in part, because so far there are no experiments considering the emergence and competition of multiple life cycles at the same time. However, these experiments have now become feasible and some experimentalists have already expressed interest.

2) Please revise the terminology (e,g, "projection matrix", "linear dynamics", and stationary state") to avoid confusion with technical math definitions.

We apologize for having used a terminology that is potentially misleading. These terms are commonly used in the field of mathematical demography, but we now switched to a more precise notation.

3) In the discussion "Similarities between models with and without group competition," please include results from models without competition as well to make the comparisons clearer for the reader.

We agree that our results are hard to understand without the background provided by that basic model. We now included a paragraph describing the results established for that model without group competition.

4) Please include additional quantitative discussions and elaborate on the effect of kernel on the outcomes.

We have now included more details on the interaction kernels.

5) Please highlight the primary takeaways of the paper among many results that are derived to underscore the main findings. The reviewers suggest that you frame these questions at the beginning of your paper to help your reader follow the modeling and results.

We agree that it is a good idea to mention the main takeaway already in the introduction and have now done so.

6) Please reorganize the Discussion section to include a recap and improve its flow for your readers.

We have rewritten the discussion along these lines.

7) Please double-check the equations for accuracy and address the concerns raised by reviewer #3.

These are concerns that we took very seriously, but we show in the revised manuscript that our previous approach was justified. For example, we can focus on the leading eigenvector as it is the only one that is biologically relevant, as all its elements are positive.

Reviewer #1 (Recommendations for the authors):I have few comments and suggestions to improve the quality of the work:1. The model is very clearly stated and a connection between current model and previous works has been established. The generalization of interaction between different groups are quite interesting and significant improvement in the modeling approach to the evolution of multicellular life cycles. However, it would be more instructive if the authors identify what form of interactions might be more relevant in the modeling, beside victim and killer. Also, it would be pedagogical if the above is compared, in a couple of sentences or a paragraph, with other frameworks in evolutionary game theory. To me, the model in the absence of interactions is a quasi-species equation where each group is a trait and every growth terms is similar to mutation to a new trait. A given life cycle is a mutational pathway in this analogy. The interaction term is interaction between the different types/traits. It would be instructive if authors, by making such connections with theoretical evolution literature, discuss what qualitative or qualitative changes they already expect from introducing the interactions kernel.

We agree that the general dynamics of a population subdivided into several classes with transitions between them by a set of differential equations does resemble the quasi-species framework. However, the interpretation of our model is very different: For example, in our model each organism passes through different stages from birth to death. In the quasi-species model, an individual maintains its identity and switching between classes occurs due to mutations. In addition to the main connection to theoretical biology in terms of demographic models, we now have added connections to other related topics.

2. The model has a complex form, inevitably. There are several mechanisms interplaying with each other. Growth rates of groups, fragmentation patterns, interaction between groups of the same life cycle, and interaction between groups belonging to different life cycles and most often the latter two are indistinguishable. Even though the model is complex, the paper is well written and the results are clearly enough stated. However, it would be very important to improve the significance of the work by highlighting a few main messages of the paper among many results that are derived to highlight the main findings. This is the part I found somewhat unclear in the current manuscript.

We have now concluded the introduction with our main question and with the key results.

3. Regarding the above comment, I want to suggest that the authors highlight, from the beginning, what questions they are trying to answer and what quantities can be calculated from the model. In principle, I expect the model says the condition for selection of a life-cycle against other ones (invasion of rare) and how it relates to model parameters. Similarly conditions for coexistence between different life cycles. Finally, the steady state values of each group abundances. These are all discussed in the paper to some extent. I just suggest highlighting the main quantities you are calculating and questions you are aiming to answer a little clearer.

Thank you. Agreed, we now spell out more clearly what these conditions mean and how they arise.

4. In the case of two competing life cycles, it is not clear, to me, what is the qualitative result. The kernel is introduced in four forms suggested by the authors (Figure 2). It is concluded that depending on whether one has a killer kernel or victim kernel one life cycle wins and in other cases there might be coexistence. Are there further results? For example can one in principle states the condition for ESS or selection advantage of one life cycle?

Unfortunately, we have no results for the fully general case. However, even the cases where the interaction kernel does not alter the competition are potentially of a lot of interest – it implies that we can derive results from exponential growth competition and they carry over to restricted systems. We now describe this more clearly.

5. (Similar to the previous comment) How does different kernels (victim or killer) improve or modify condition for selection of one life cycle while competing with another? The answer is detailed but it is useful to sum up the overall behavior cycles.

Thank you for this question. We reworked the results structure and put the presentation of killer and victim kernels to the dedicated subsection 3.5. There, we show that killer and victim kernels are the special cases, where the result of evolution is a single life cycle dominating others. For a killer kernel, the winning life cycle is the one with the largest growth rate. For a victim kernel, the winning life cycle is the one with the largest carrying capacity.

6. I want to comment that the three life cycle competition results was interesting and the analogy with rock-paper-scissor game was clarifying the findings.

We agree that it is interesting, but it is not very frequent. Now we also refer to the ecological literature on non-transitive interactions, which is another way to phrase this.

7. Does the introduction of kernel term overall promotes coexistence between two life cycles?

No, not necessarily, as for many kernels there is no coexistence. But without such a kernel, there is no generic scenario for coexistences in the first place.

8. While the paper has a theory approach and the generality of results justifies it, it would be very useful to connect the general finding with the observations in experimental evolution of multicellularity. For example, discuss cases where there are unique life cycles and when there are more than one.

Unfortunately, the experimental literature on the evolution of life cycles is only about to take off. So far, we only know that some experimental scientists start to work on experiments in this direction, but preliminary results from Will Ratcliff lab in Georgia Tech indicate that evolution of colonial yeast may lead to coexistence of two life cycles with different colony sizes -- a scenario predicted by our model.

The evolution of multicellular life cycles is a central process in the course of the emergence of multicellularity. The model suggested by authors connects evolution of multicellular life cycles to evolutionary game theory. The introduction of the interaction terms seems to be a great modeling way to discuss under what circumstances different life cycles can coexist or when one life cycle is chosen among other potential ones due to a natural selection among life cycles. The results are discussed in some details but due to complexity in some cases examples are used. I recommend this work for publication in Journal eLife after revision. It is a new model with high impact on the field of evolution of multicellular structures.

Thank you.

Reviewer #2 (Recommendations for the authors):1) The first paragraph does not contain any references to back up the claims presented by the authors. It would be good to address this and add appropriate references.

Thanks, now we added appropriate references.

2) In Figure 1 it would be good to explicitly state in the caption what the different colors for the groups mean.

Done. Here, colors code for number of cells in an organism. Thanks.

3) In the discussion "Similarities between models with and without group competition "it would be good to explicitly refer to results from results from models without competition (which I think are only seen in Figure 2A) to make the comparisons clearer for the reader.

We understand that the paper was very hard to read without an explicit discussion of these cases, which we have now added.

Reviewer #3 (Recommendations for the authors):Please find my detailed recommendations for the authors belowPage 7, line 142: “The projection matrix $A$…” It is not clear to me what the authors mean by “projection matrix”. From a mathematical point of view, a square matrix $P$ is called a projection matrix if it is equal to its square, i.e. if $P^2=P$. However, it does not seem that in general $A^2 = A$.

True. Thank you for pointing out this source of possible misunderstanding. We took this term from the literature of formal demography, where the word “projection” uses the meaning of extrapolation into future. Now, we use more precise term “population projection matrix” and added additional explanation where it is introduced.

Page 10, line 219: “Unlike the resident dynamics, the dynamics of the invader population is linear…” This dynamics is linear only if $x^{(R)}$ is constant (e.g., at equilibrium). Therefore, it would be helpful to use different notations for the function $x^{(R)}(t)$ and its value at the equilibrium $x^{(R,*)}$ in formulas 11-12 describing these dynamics.

Agreed, it also makes the notation in Equations 11 and 12 consistent with the rest of the paper.

Page 11, lines 245-246: “Numerical simulations show that an isolated life cycle always comes to the same stationary state $x^*$ from any initial distribution of group sizes.” Could you please add more details about these numerical simulations? What kind of life cycles were considered? What birth and death rates have been chosen? How was the competition matrix generated?

We added an illustration of how different initial states converge to the same steady state (Figure 2B,C) for a single life cycle and a chosen set of birth/death/interactions rates. We also additionally checked the convergence to the same steady state with the dataset obtained for Figure 4B – with 7 competing life cycles, 13000 randomly sampled sets of rates (all drawn independently from exponential distribution with unit rate parameter), and 100 replicates of evolution dynamics for each set different by the initial conditions. 75% of 13000 sets featured a dominance of a single life cycle. In all these cases, the variance in the final number of groups across replicates of a set was less than 10^(-4). Thus, which a high accuracy, the dynamics of a single life cycle comes to the same stationary state.

Throughout the text, the authors discuss the results on various special types of a competition matrix (constant, killer kernel, victim kernel). These special cases are used to obtain analytical conclusions that cannot be drawn in a general case. This helps readers deepen their intuition about this model. However, I wonder if there are any real biological systems that can be described by these special kernels? If so, it would be very helpful to include the corresponding discussion in the text.

We speculate about possible forms of empirical interaction matrices in the discussion, but so far the experimental field does not look into such interaction kernels. But we hope that this kind of data is available in a few years!

Page 25, line 515: “In the linear model, the stationary state is an exponentially growing population…” From a mathematical point of view, a stationary state of a system is a state with all observables independent of time. Therefore, I am not sure if it is appropriate to use the term “stationary state” here.

Agreed, we should use steady state here instead (note that demographers speak of stable demographies when populations are growing exponentially in size, so it depends on the field which words seem to be appropriate to characterize such dynamics).

Page 25, lines 526-529. It is not clear what does the index $s$ mean in the corresponding formulas.

Thanks, this was a typo left from the earlier draft. Index $s$ does not represent anything and is removed.

Appendix A1. As was shown in the previous work, the solution to Equation (14) depends on the leading eigenvalue $\λ^*$ of the matrix $A$ and the corresponding eigenvector $w^*$. After refreshing that, the authors showed that a stationary state of the dynamics governed by Formula (21) (i.e., by Equation (14) in the case of the killer kernel) depends on $\λ^*$ and $w^*$ (see Formula (22)). This is correct. However, for each eigenvalue $\λ$ and each corresponding eigenvector $w$, Formula (22) produces a stationary state of the dynamics governed by Formula (21). It is not clear to me why, in the case of the killer kernel, the authors consider only the leading eigenvector? If such a conclusion is made on the basis of the above results for the linear case, please explain why these results can be generalized to the killer kernel.

Right, any eigenvector of the population projection matrix can work in this equation. However, only the eigenvector associated with the leading eigenvalue has positive-only elements. This follows from Perron-Frobenius theorem (we added a paragraph explaining why this theorem applies to our matrices). All other eigenvectors contain negative elements, which would correspond to negative number of groups at the stationary state, so these would not be a biologically meaningful solutions.

And the last question. Is it possible that the leading eigenvalue $\λ$ has an eigenspace of dimension higher than 1 so that stationary states (22) form a “line” of equilibria?

No, such a situation is forbidden by the Perron-Frobenius theorem as well.

Page 25, line 528. I guess, it should be $x_{s,j}^*$ instead of $x_{s,j}$.

Yes, thanks!

Page 27, line 551-552. What is $N_0$? How is it related to $N^*$?

Thank you, this was a typo – it refers to $N^*$!

Section 3.2. The authors state that the dynamics are very complex in the case of competition between multiple life cycles, and therefore consider these dynamics only in some special cases. I agree that the dynamics are complex. However, as a reader, I have no intuition about how the model works in the general case, which is the most interesting question for me. Therefore, it would be helpful to add some numerical simulations exploring the above dynamics in the general case. It would also be useful to present some statistics illustrating the average number of different life cycles presented in a stationary state as well as the number of extinct ones. Maybe some life cycles are rarely observed in a stationary state, while others are widespread under a broad range of parameters and initial conditions?

Right, we did not have this intuition either and the question you raise is an interesting one! We now added new section 3.3. considering the competition between all life cycles with groups smaller than 4 cells (there are 7 such life cycles). We classified the stationary states: most often we find a dominance of a single life cycle, like in a linear model. However, in rare cases, we observe quite complex outcomes – e.g. bi-stability between different combinations of coexisting life cycles.

Page 16, Figure 3D. If possible, could you please mark in different colours trajectories approaching different stationary states?

Thank you, that is a very good idea!

Lines 450-451. “Yet, it is possible to introduce a linear transformation $x \rightarrow Cy$, where $C$ is a matrix, which will make the linear term in our model diagonal.” Where is a proof that the matrix $A$ can be diagonalized?

We have no such proof but it is not needed. Here we say that even if $A$ can be diagonalized, then our system is still not equivalent to generalized Lotka-Volterra due to the complications occurring with competition term. If $A$ cannot be diagonalized, as you suggested, then our system is not equivalent to GLV for sure, and our message still holds. In the revision, we clarified this sentence.

I think the structure of the Discussion section could be improved. For example, I do not understand why you put paragraph 2 (lines 431-442) at the beginning of the Discussion? This paragraph is about a generalization of your model and its results, and it is strange to discuss the generalization of the results before discussing the results themselves. Second, it might be helpful to add a brief recap of the problem under study, and a brief overview of the model at the beginning of the Discussion section before discussing the position of your study in various contexts, and related results (lines 443-505).

Thank you, we have now rewritten the discussion along the suggestions of you and the other reviewers.